# Socio-hydrologic perspectives of the co-evolution of humans and groundwater in Cangzhou, North China Plain

Songjun Han[1, 3], Fuqiang Tian[2], Ye Liu[2, 4], Xianhui Duan[5]

[1]State Key Laboratory of Simulation and Regulation of Water Cycle in River Basin, China Institute of Water Resources and Hydropower Research, Beijing 100038, China;
[2]Department of Hydraulic Engineering, State Key Laboratory of Hydro-science and Engineering, Tsinghua University, Beijing 100084, China;
[3] National Center of Efficient Irrigation Engineering and Technology Research, Beijing 100048, China;
[4] Social Development Department, National Development and Reform Commission, China, 100824
[5]Hydrology and Water Resources Survey Bureau of Cangzhou, Cangzhou, 061000, China.

*Correspondence to:* Songjun Han (hansj@iwhr.com); Fuqiang Tian (tianfq@tsinghua.edu.cn)

**Abstract.** This paper presents a historical analysis from socio-hydrologic perspectives of the coupled human–groundwater system of the Cangzhou region in the North China Plain. The history of the "pendulum swing" for water allocation between the economic development and aquifer environmental health of the system is divided into five eras (i.e., natural, exploitation, degradation and restoration, drought-triggered deterioration, and returning to the balance). The system evolution was interpreted using the Taiji-Tire model. Over-exploitation was considered as the main cause of aquifer depletion and the groundwater utilization pattern was affected by the varying groundwater table. The aquifer depletion enhanced the community sensitivity of humans toward environmental issues, and upgraded the social productive force for restoration. The evolution of the system was substantially impacted by two droughts. The drought in 1965 induced the system from natural condition to groundwater exploiting. The drought from 1997 to 2002 resulted a pulse in further groundwater abstraction and dramatic aquifer deterioration, and the community sensitivity increased rapidly and induced the social productive force to a tipping point. From then on, the system is returning the balance through new policies and water-saving technologies. Along with the establishment of a strict water resource management strategy and the launch of the South-to-North Water Diversion Project, further restorations of groundwater environment would be implemented. However, a comprehensive and coordinated drought management plan should be devised to avoid the irreversible change of the system.

# 1 Introduction

Through adaption processes, humans co-evolve with the hydrological system, resulting in a "pendulum swing" in the balance point of the human–water system (Kandasamy et al., 2014; Sivapalan, 2015). Kandasamy et al. (2014) first characterized the concept of the "pendulum swing" by tracing 100-year history of the competition for water between agricultural development and environmental health in the Murrumbidgee River Basin. Similar dynamics were also found in the human–water system in the arid Tarim River Basin (Liu et al., 2014) and in human–flood interactions (Baldassarre et al., 2015). A pendulum swing can be divided into four typical stages: (1) the initial exploitation stage, which is focused exclusively on economic development; (2) the onset environmental degradation stage, which is accompanied by the introduction of remedial infrastructure; (3) the widespread environmental degradation stage, which leads to the necessity of mitigation measures; and (4) the recovery stage, at which ultimate solutions are implemented. The pendulum swing of a human–water system can be clarified by considering all interactions within a universal socio-hydrologic framework (Kandasamy et al., 2014; Liu et al., 2014). The importance of socio-hydrology has been recognized by the International Association of Hydrological Sciences through their new "Scientific Decade" (2013–2022) entitled "Panta Rhei (Everything flows)" (Montanari et al., 2013), which aims "to reach an improved interpretation of the processes governing the water cycle by focusing on their changing dynamics in connection with rapidly changing human systems."

A socio-hydrological system contains human, hydrological, and environmental sub-systems. The Taiji-Tire model proposed by Liu et al. (2014) was first used as a framework for elucidating the complexities of socio-hydrological systems that co-evolve with the direct or indirect interactions between factors from both human and water perspectives. In the model, a Taiji symbol, which is a term originating from a unique concept in Chinese philosophy, is used to describe the direct human–water relationship in a specific socio-hydrological system, whereas a human–water tire is used to represent the indirect effect of external natural and social factors affecting the system. The evolution of a socio-hydrological system is thought to be driven by the interactions between two main factors, namely, natural variability and social productive force (Liu et al., 2014). Social productive force refers to the combination of all factors that enable humans to utilize resources and create better material and spiritual products.

During the recovery stage of a pendulum swing evolution, environmental protection actions are usually conducted because of a high social awareness for environmental risk or welfare (Di Baldassarre et al., 2013). Elshafei et al. (2014) proposed a new concept of community sensitivity, which refers to the sensitivity of humans to the changing environment, to signify the social awareness for environmental welfare. A high community sensitivity implies that humans feel the pressure of environmental deterioration, motivating them to restrain human activities to restore environmental health. The concept of community sensitivity was also used to analyze the switching of the support between flood protection and wetland preservation in the Kissimmee River Basin, Florida (Chen et al., 2016). It can be incorporated into the Taiji-Tire model to improve its explanatory power on a specific human–water system.

Being a ubiquitous and stable source of high-quality fresh water, the groundwater system is closely connected with human. Driven by the increasing demand for water resources, as well as the need for combating drought, improper exploitation of groundwater resources has resulted in serious environmental crises, particularly in regions with primarily groundwater-fed irrigation (Taylor et al., 2013) including North China Plain (NCP) (Liu et al., 2001; Chen et al., 2003;

Zheng et al., 2010). North China Plain is an important agricultural region of China. The main crops in this region are wheat and maize. Irrigation is necessary to maintain high levels of grain production (Liu et al., 2001), and 70% of the cultivated land is irrigated, consuming 70% of the total water supply. Given that over-exploitation of surface water causes the lower reaches of streams to dry up commonly in North China, groundwater has become the regular supply for irrigation since the 1970s. The density of pumped wells have led to severe depletions of both unconfined and confined aquifers (Zhang et al.,

1997; Zheng et al., 2010). Quite a few studies have been carried out to better understand the spatiotemporal variations in groundwater depletion across the NCP and to develop sustainable groundwater management options (Cao et al., 2013; Liu et al., 2008); in these works, the effects of human-induced change were evaluated by a scenario-based method (Wang et al., 2008; Liu et al., 2011; Liu et al., 2008). However, two-way feedback should be considered for a better understanding of coupled human–groundwater systems (Montanari et al., 2013).

A representative example of the co-evolution of the human–groundwater system is in Cangzhou in the northeastern coastal plain of the NCP, which has the most serious depression cone since the 1970s (Liu et al., 2001). The balance between groundwater extraction and efforts to mitigate and reverse the consequent degradation of the aquifer has evolved in Cangzhou since 1949 (Li et al., 2013). In recent years, aquifers have been recovered through the implementation of several strict measures (Han et al., 2013). The history of the co-evolution of the human–groundwater system in Cangzhou seems like

a "pendulum swing," particularly as to how the groundwater crisis unfolded and how it was addressed. Therefore, Cangzhou is selected as the study area in this paper.

The objectives of this paper are as follows: (1) to chart the history of the groundwater utilization in Cangzhou, focusing on the dynamics of the human–groundwater interactions that lead to the "pendulum swing" in the balance point in groundwater allocations between humans and aquifer ecosystems, as well as the natural variability and social factors that

contributed to it; (2) to use the Taiji-Tire model to interpret the interactions and co-evolutions of the human–groundwater system in Cangzhou. The Taiji-Tire model will be specific to the groundwater system and will be incorporated with the concepts of community sensitivity.

## 2 Study area, data, and methods

### 2.1 Study area

Groundwater pumping from aquifers has substantially increased since the mid-1960s. Thus, the NCP aquifer system has become one of the most over-exploited aquifers in the world (Kendy et al., 2007; Liu et al., 2008). Cangzhou region (total area: 14,056 km$^2$) is located in the east of the NCP and in the downstream of the Haihe River Basin (Fig. 1). Cangzhou is a

prefecture-level city of Hebei Province consisting of 4 county-level cities and 10 counties. Cangzhou Municipal Government abides by the provincial and national policies of water resource management, and it devises policies for the entire region. The Cangzhou Water Resources Bureau, which is a department of Cangzhou Municipal Government, is responsible for water resource affairs and guides the Water Resources Bureau of the 10 counties and 4 county-level cities. In turn, this

5   agency is guided by the Hebei Water Resources Department, the Haihe River Water Resources Commission, and the Ministry of Water Resources. Cangzhou is characterized by strong inter-annual variable precipitation and occasional extreme droughts. In 2013, the total population of Cangzhou was $7.34 \times 10^6$. The cultivated land area is 8,066 km$^2$, of which 5,424 km$^2$ is irrigated. Given that surface water is intercepted by reservoirs in the upstream, natural streams have nearly dried up, leaving groundwater as the main water source for irrigation.

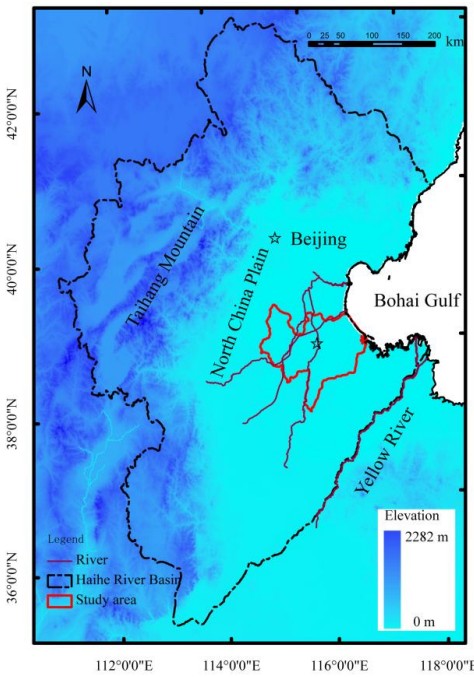

**Figure 1. Location of Cangzhou in the North China Plain**

       The groundwater resides in aquifers of porous quaternary alluvial deposits, which can be divided into four major aquifer layers (I–IV). The thickness of each layer ranges between 20 m and 350 m. Aquifer Layer I is unconfined, and infiltration from precipitation is the main recharge source. The other three layers are confined. Saltwater accounts for 98% of Aquifer

15  Layer II, with low exploitation capacity. Pumping wells mainly extract fresh groundwater from Aquifer Layers I and III. In this study, Aquifer Layers I and III are referred to as "shallow aquifer" and "deep aquifer," respectively. Groundwater levels have steadily declined because of over-pumping, resulting in saltwater intrusion and land surface subsiding (Kendy et al., 2003). In 2013, the groundwater withdrawal from the shallow and deep aquifers were $259 \times 10^6$ and $743 \times 10^6$ m$^3$, respectively. In recent years, several strict measures have been implemented for sustainable groundwater management to mitigate aquifer

20  depletion (Han et al., 2013)**.**

## 2.2 Data and methods

The study is performed based on the data of major events, policies and initiatives on the social development, aquifers, and natural environment related to the groundwater system. The statistical analysis is based on the quantitative data of groundwater and surface water hydrology, climate, agricultural infrastructures and groundwater utilizations, as well as the population and GDP from 1949 to 2015. The data before 1985 was acquired from the Water Resource Annals of Cangzhou (Xue, 1994), and the data after 1985 was obtained from the Hebei Rural Statistics Yearbook, the Cangzhou Water Resources Bureau, and the Cangzhou Hydrology and Water Resources Investigation Bureau. As presented in Fig. 2, the time series of the groundwater withdrawal from both the shallow and deep aquifers were used to detect the behavior of water users. The average water table depth of the shallow aquifer and the depth of the depletion cone of the deep aquifer were used as the main indices for the groundwater system. The changes of annual precipitation and surface water inflow were used to analyze the natural variability. The time series of number of wells and irrigated area (including water-saving irrigated area) were analyzed to detect the changes in infrastructure, along with the social development which were revealed by the changes in population, and grain production.

The history of the pendulum swing of the human–groundwater system is traced through a review of the literature on the evolution of the human-groundwater system. Five distinct eras, a pre-era and four eras following the typical narrative of a pendulum swing (exploitation, degradation and restoration, drought-triggered deterioration, and returning to the balance), are divided by distinguishing the events that can be regarded as tipping points. There are two criteria to judge the tipping points: (1) groundwater utilization pattern is significantly different; (2) groundwater table of the shallow and (or) deep aquifers is significantly changed before and after the events. In addition, the division is also validated through the statistical analysis of the quantitative data of the groundwater table depth and groundwater utilizations. The events in terms of the relative emphasis placed on social development, that significantly affect the groundwater utilization or aquifers environmental health are chosen to conduct the narrative (Table 1).

The drought occurred in 1965, which triggered the groundwater utilization in Cangzhou, was chosen as the beginning of the exploitation era. The period before is regarded as the pre-era, of which the human-groundwater system was dominated by natural variability. During the exploitation era, the groundwater withdrawal began to increase, and the water table began to decline. But the trends ceased temporarily in 1983, when the Cangzhou Hydraulic Engineering Society submitted an appeal to government for integrated management of groundwater resources. After that, comprehensive management measures were implemented to address the groundwater crisis, and the deterioration was mitigated to a certain extent. Therefore, this event is chosen as the beginning of Era 3. The increasing of groundwater utilization and the declines of water table began to slow down during the Era 3. However, the mitigation was terminated by a persistent drought lasted from 1997 to 2002, which is regarded as Era 4, when a pulse in groundwater utilization and aquifer depletion occurred. After the drought, the system has been returning to the balance with decreasing groundwater utilization and rising water table, which is regarded as Era 5.

**Table 1. Major environmental and societal events related to human-groundwater system in Cangzhou (bold sentences are dividing events for different eras)**

| | Environment | Policy |
|---|---|---|
| Era 2 (1965 - 1982) | ◆ **1965: serious drought occurred in North China**<br>◆ 1969: depression cone of the deep aquifer first occurred in Cangzhou City<br>◆ 1970: land subsidence first occurred in Cangzhou City | ◆ 1966: well drilling was specified as an important measure to combat drought in North China by the State Council, and Cangzhou government began to promote construction of motor-pumped wells for groundwater exploitation.<br>◆ 1970: State government began to fund motor-pumped wells construction for food production in Northern China<br>◆ 1973: the headquarters for the motor-pumped wells construction were established in Cangzhou<br>◆ 1979: Cangzhou Hydraulic Engineering Society was set up |
| Era 3 (1983 - 1996) | ◆ 1995:the cumulative subsidence in Cangzhou City reached 1443 mm | ◆ **1983: Cangzhou Hydraulic Engineering Society submitted an appeal on water resources utilization to the government**<br>◆ 1985: Hebei Provincial Government issued Regulation of Water Resources<br>◆ 1985: Cangzhou Water Conservancy Bureau issued suggestions for strengthening the management of groundwater resources<br>◆ 1993: Surface water begun to be diverted from the Yellow River to Cangzhou |
| Era 4 (1997 - 2002) | ◆ **1997-2002: Persistent drought occurred in North China**<br>◆ 2001: water table at the center of the depression cone was deeper than 100 m<br>◆ 2001: the cumulative subsidence in Cangzhou City was 2,236 mm | ◆ 1997: well drilling was regarded as the immediate strategy for addressing the drought in the emergency meeting of Hebei<br>◆ 1997: Irrigation and Drainage Group of Hebei Hydraulic Engineering Society appealed the necessity of water saving<br>◆ 1998: Hebei Government issued several provisions of water saving for the whole society<br>◆ 1999: Water-saving irrigation was identified as the focus of water conservancy constructions in Cangzhou<br>◆ 2002: Hebei Provincial Government specified the over-exploited regions and the severely over-exploited regions in the plain area, and issued to further strengthen the groundwater resources management<br>◆ 2002: the Disaster Prevention and Mitigation Group of Hebei Hydraulic Engineering Society suggested to change the idea of drought relief |
| Era 5 (2003 - ) | ◆ 2007: water table depth at the center of the depression cone rose back to 86 m beneath the ground<br>◆ 2014: the latest specified over-exploited regions and severely over-exploited regions of Cangzhou shrinked | ◆ **2003: Ministry of Water Resources issued the Suggestions Concerning the Strengthening of the Water Resources Management of Groundwater Over-Exploitation Regions**<br>◆ 2004: Cangzhou Ground subsidence prevention and control work leading group was set up<br>◆ 2005: Cangzhou Government issued the decision on shutting down wells lack of management in cities<br>◆ 2012: Cangzhou Government launched the implementation schemes of the strictest water resource management strategy<br>◆ 2012: Cangzhou Government issued the high efficiency water-saving irrigation planning<br>◆ 2015: Hebei Government issued the Regulations on groundwater management of Hebei<br>◆ 2015: the South-to-North Water Diversion Project was established |

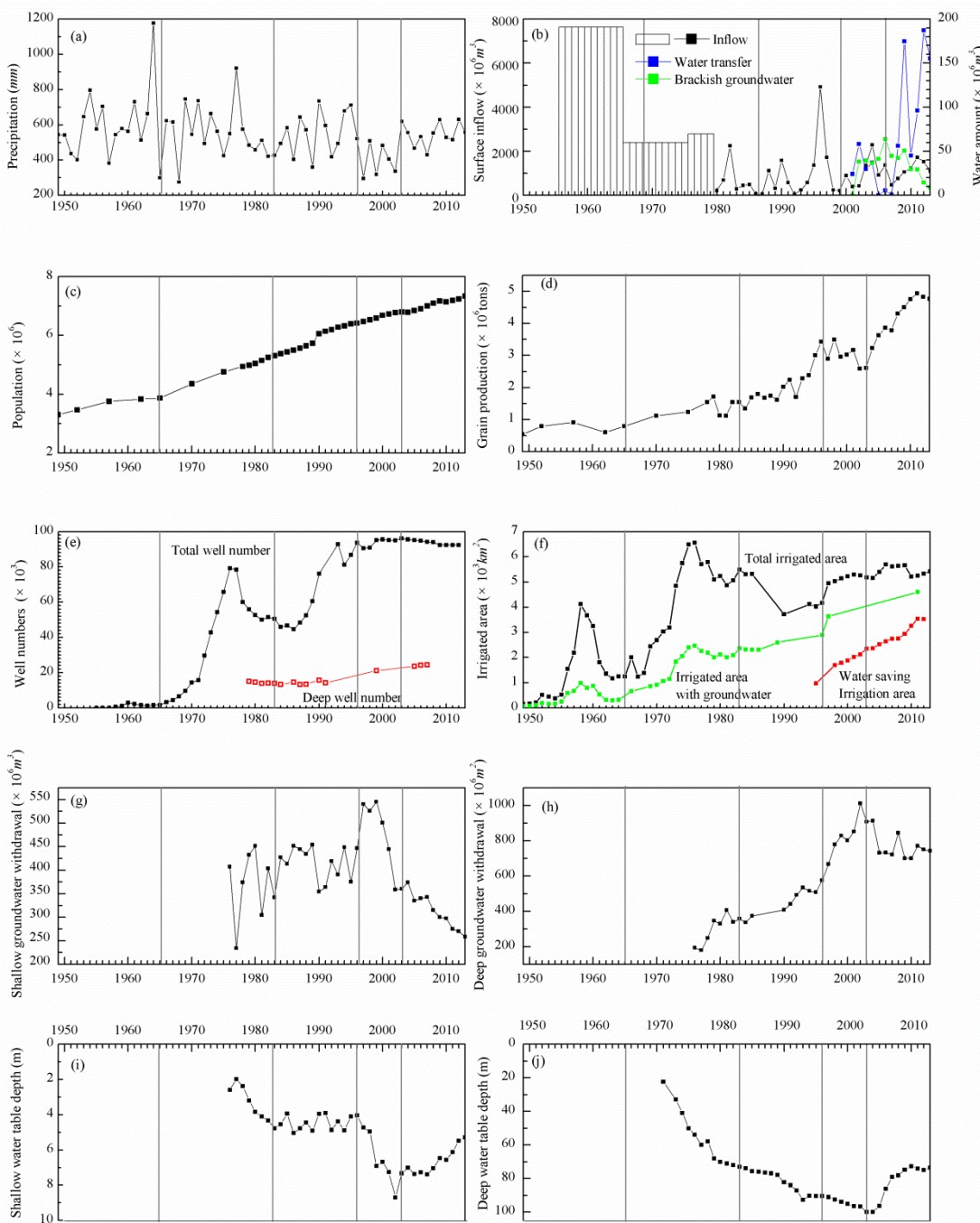

**Figure 2. Time series of (a) precipitation, (b) surface inflow (before 1980 the data is the average value), (c) population, (d) grain production, (e) well numbers (red line is referred the number of deep well), (f) irrigated area, (g) shallow groundwater withdrawal, (h) deep groundwater withdrawal, (i) shallow water table depth, and (j) water table depth of the depletion cone in Cangzhou during the study period (1949–2013)**

The community sensitivity of the human-groundwater system is specified to the social attitude against groundwater exploitation and social awareness on the aquifers environmental crises. In Cangzhou, as well as in Hebei, community concerns about water resource crisis are represented by, and always conveyed to the policy makers through the Hydraulic Engineering Society mainly because of two reasons. First, the Hydraulic Engineering Society is the largest non-governmental organization related with the affairs of water resources in corresponding region, and aims to promote sustainable water resources management. Besides, the Hydraulic Engineering Society has the most diverse membership, including individuals, enterprises, public institution, colleges and other stakeholders. Groundwater issues are among the most concerned ones of water resources management in Cangzhou. In order to detect the changes of the community sensitivity, the attitude and awareness toward groundwater issues of the Cangzhou Hydraulic Engineering Society, and the Hebei Hydraulic Engineering Society are carefully reviewed.

Under the framework of the Taiji-Tire model, the Taiji is specified to the direct interactions between the groundwater utilization and the aquifers in Cangzhou. The status of the shallow aquifer is represented by the water table depth, while the status of the deep aquifer is represented by the depth of the depression cone center. The influences of humans on groundwater are detected from the relationship between changes in shallow groundwater table and the water withdraw, and the co-evolution of the depth of the depression cone center of deep aquifer with water withdrawal from 1976 to 2013. The feedbacks of changes in groundwater sub-system on humans were detected from the relationship between groundwater use regime and the shallow water table depth. The outer Tire represents all of the social and natural factors that indirectly influence the human–groundwater system. The time series of annual precipitation and surface inflow were used to represent the natural variability in Cangzhou, and its impacts on the groundwater sub-system and human sub-system were detected from the changes in corresponding groundwater table depth and withdrawals, respectively. The impacts of society were analyzed through the changes of the emphasis level of the social productive force, which is detected from the changes in the number of wells, in irrigated areas with groundwater, and in policies for groundwater exploitation. In this study, we regarded social restorative force as a subtype of social productive force, which can be detected from the changes in water-saving irrigation areas and in policies for creating water-saving technologies.

**3 Pendulum swing in groundwater utilization**

The five distinct eras are:

Era 1 (Pre–1964): Natural variability dominates the human–groundwater system

Era 2 (1965–1982): Expansion of groundwater exploitation and onset of aquifer depletion

Era 3 (1983–1996): Awareness of environmental degradation and attempts for restoration

Era 4 (1997–2002): Drought-triggered pulse in groundwater abstraction and aquifer depletion

Era 5 (2003–present): Returning to the balance

### 3.1 Era 1 (Pre-1964): Natural variability dominates the human–groundwater system

The groundwater resources utilization in Cangzhou has a long history. Archaeological discoveries show that Cangzhou residents drilled wells to obtain drinking water as early as in the Han Dynasty (approximately 220 BC–220 AD). Historical records indicate that groundwater exploitation for irrigation in Cangzhou can be traced back to 1266, which was over 700 years ago. In 1949, the irrigated area with groundwater was only 74 km$^2$, which was distributed in fewer places where shallow freshwater resources were abundant. From the early 1950s to the mid-1960s, the Haihe River basin was rich in surface water resources. The emergence of serious salinization problems necessitated the establishment of a drainage-oriented policy for lowlands in Cangzhou. Many reservoirs and diversion projects were constructed, reducing the need for groundwater resources. Given that most wells were made of bricks and earth, groundwater utilization was restricted by the lack of infrastructure. By 1964, only a small portion of the wells (1,524) were pumped by motors, and the irrigated area with groundwater was 321 km$^2$, which only accounted for 2.3% of the total area of Cangzhou. Therefore, this era was characterized by small-scale groundwater utilization because of less groundwater demand as well as technological incapability. The interaction between humans and the groundwater was weak. Humans were insensitive to groundwater change, and the groundwater sub-system was unaffected by humans at a large scale.

### 3.2 Era 2 (1965–1982): Expansion of groundwater exploitation and onset of aquifer depletion

The beginning of Era 2 was an environmental event, a catastrophic drought occurred in 1965. The drought significantly threatened food production in North China. In 1966, the State Council organized a conference to combat drought in North China, and they specified well drilling as an important measure. Numerous well-drilling teams were organized after hydrogeological investigations in the NCP. In response, the Cangzhou government began to promote the construction of motor-pumped wells. In 1970, the number of motor-pumped wells increased to 14,328 from 1,548 in 1965, and the irrigated area with groundwater increased to 920 km$^2$. After the drought, the benefits of the emergency wells encouraged the continued construction of groundwater exploitation infrastructures.. In August 1970, the state government held a conference on agricultural production in Northern China and decided to accelerate the agricultural development in regions with food shortage, including Cangzhou. Specifically, the construction of motor-pumped wells was included in the national plan. The emergency measure of groundwater exploitation became a normal one. From 1970 to the early 1980s, motor-pumped well constructions were generally supported by national special funds. In 1973, the headquarters for the construction of motor-pumped wells were established in Cangzhou. After 1975, when the irrigated area with surface water reached a maximum of 4,084 km$^2$ (the area irrigated with groundwater was 2,401 km$^2$ in the same year), surface water resources were gradually exhausted (which can be detected from the changes in runoff of two rivers shown in Fig. 2(b)). Groundwater then became an important water resource for irrigation. In 1982, the number of motor-pumped wells reached 51,611, and the irrigated area with groundwater reached 2,086 km$^2$ (Figure 2(e)), which is 41.3% of the total irrigated area. Benefit from the groundwater utilization, the grain yield in this region increased from $0.79 \times 10^6$ tons in 1965 to $1.54 \times 10^6$ tons in 1982.

**Table 2. Changes in shallow and deep groundwater withdrawal and table depth from Era 2 to Era 5**

| | Shallow groundwater | | | Deep groundwater | | |
|---|---|---|---|---|---|---|
| | Withdrawal $\times 10^6 m^3$ | Depth m | Trends m/yr | Withdrawal $\times 10^6 m^3$ | Depth m | Trends m/yr |
| Era 2* | 372.4 | 3.20 | -0.39 | 291.1 | 64.87 | -3.11 |
| Era 3 | 411.9 | 4.47 | 0.03 | 436.4 | 82.22 | -1.63 |
| Era 4 | 485.9 | 6.54 | -0.75 | 822.7 | 96.72 | -2.47 |
| Era 5 | 315.3 | 6.66 | 0.2 | 774.0 | 82.69 | 3.01 |

* According to the data from 1976 to 1982.

The rapid increase in groundwater exploitation drastically deepened the groundwater table of both the shallow and deep aquifers. The average annual shallow groundwater withdrawal from 1976 to 1982 was $372.4 \times 10^6$ m$^3$ (Table 2). In 1982, the regional average shallow groundwater table declined to 4.32 m beneath the ground, with an annual decline of 0.39 m from 1976 to 1982. As a result, many shallow wells were abandoned because of the significant decline in the groundwater table. Motor-pumped wells had to be drilled with increasing depth, signaling the start of a vicious cycle. Groundwater exploitation had to be drilled deeper than usual, and the  deep groundwater withdrawal significantly increased, from $193 \times 10^6$ m$^3$ in 1976 to $357.7 \times 10^6$ m$^3$ in 1983, which exceeded the sustainable volume ($292 \times 10^6$ m$^3$ according to Wang and Wang (2007)) since 1979. With increasing exploitation, the water table of the deep aquifer rapidly declined. The depression cone of the deep aquifer in Cangzhou first appeared in 1967, whereas the depth of the water table at the center of the depression cone was 22.5 m beneath the ground in June 1971. By 1982, the water level at the center of the depression cone was as deep as 72 m, with an annual decline of 3.11 m from 1973 to 1982. The drilling depth increased because of the drop in deep groundwater table, and the cost for both well drilling and installation of new motor-pumped wells rose. In addition, several environmental problems were triggered. Land subsidence occurred in Cangzhou City in 1970 because of the presence of the depression cone. The cumulative volume of subsidence in 1970 was 9 mm, and it continued to be worsen. Because of the formation of the depression cone in Aquifer Layer III, the natural recharge–discharge balance was destroyed between the fresh water originally in Aquifer Layer III and the saltwater in the overlying aquifer layer II. The leakage recharge of saltwater from Aquifer Layer II to Layer III increased, leading to saltwater intrusion.

**3.3 Era 3 (1983–1996): Awareness of environmental degradation and attempts for restoration**

The beginning of Era 3 was a societal event, which represented the upgrading of community sensitivity. The intensification of the groundwater crisis raised the public clamor to address the water crisis. In 1983, the Cangzhou Hydraulic Engineering Society, which was set up in 1979 as the response to public concerns on water problems, submitted an appeal document entitled "Appeal for the Rational Exploitation of Water Resources" to the government. In this appeal, proposals for comprehensive water resource management were proposed. The appeal revealed that humans felt the pressure of the aquifer deterioration, and can be regarded as an increasing community sensitivity. As a response to the increasing community sensitivity, the Hebei Provincial Government enacted the Regulation of Water Resources in 1985,, which stated

that the exploitable volume of groundwater in urban areas should be strictly controlled, and the exploitation of groundwater in rural areas should be reasonably planned. In June 1985, the Cangzhou Water Conservancy Bureau issued regulations for strengthening the management of groundwater resources in which comprehensive water resource management measures were detailed. Specifically, shallow groundwater exploitation should be prioritized, deep groundwater exploitation should be restricted, and brackish water should be reasonably utilized. Furthermore, a licensing system for well drilling was established, and the planting of crops, such as rice, which consumes large amounts of water, was forbidden. Since the 1980s, well drilling was no longer subsidized by the central government, resulting in a sharp decrease in the number of wells. To fill the gaps, surface water was diverted from the Yellow River in 1993. According to the Water Law of China released in 1988, and the Implementation Measures of the Licensing System for Water Taking issued in 1993, in Cangzhou where groundwater resources are over-exploited, groundwater exploitation should be strictly controlled, and the expansion of the exploitation is prohibited.

The aforementioned measures relied on comprehensive water resource management, by which deep, medium, and shallow groundwater exploitations were governed by unified planning, with shallow aquifer exploitations given priority. Consequently, the increasing deterioration of groundwater resources in Cangzhou was halted. From 1984 to 1996, the shallow groundwater withdrawal slightly decreased (by approximately $2 \times 10^6$ m$^3$ per year), whereas the average shallow groundwater table rose at the rate of 0.02 m per year (Table 2). The increase in the deep groundwater withdrawal slowed down (by approximately $18.8 \times 10^6$ m$^3$ per year). Consequently, the water level at the center of the depression cone declined slowly at an annual rate of 1.63 m. It should be noted that groundwater level was still going down, indicating an inadequacy of community awareness.

**3.4 Era 4 (1997–2002): Drought-triggered pulse in groundwater abstraction and aquifer depletion**

Unfortunately, the positive trend ceased after the outbreak of a serious drought in 1997, which started the Era 4. The precipitation in Cangzhou was only 296.3 mm (53.9% of the mean annual value) in 1997, and the average annual precipitation was only 391.7 mm from 1997 to 2002 (Table 3). However, because surface water has already been exhausted since the 1980s, well drilling was taken to be the immediate strategy for addressing the drought, which was confirmed during the emergency meeting of Hebei Province. During this period, the annual average shallow groundwater withdrawal was $485.9 \times 10^6$m$^3$, which represented an increase by 16.5% compared with $417.2 \times 10^6$m$^3$ during 1984–1996, and the annual average deep groundwater withdrawal rapidly increased from $455.2 \times 10^6$ m$^3$ during 1984–1996 to $822.7 \times 10^6$m$^3$, which represented an increase by 80.7%. Accordingly, the shallow groundwater level rapidly declined again, from 4.03 m depth beneath the ground in 1996 to 8.69 m in 2002, with an annual decline of 0.75 m. At the same time, the water table at the center of the depression cone rapidly declined again from 90.4 m depth beneath the ground in 1996 to 111.1 m in 2001 (the deepest value), with an annual decline of 2.47 m. The area with the water table of the Aquifer Layer III deeper than 80 m beneath the ground dramatically increased from 157 km$^2$ in 1996 to 421 km$^2$ in 2002. In view of the sharp decline in the groundwater table, the environment deteriorated. The cumulative subsidence to 2001 was 2,236 mm, with a rate of 100.5

mm/a. The areas with subsidence larger than 500 and 800 mm are 9,717 and 3042 $km^2$, respectively, which account for 92.9% and 29.1% of the total area of Cangzhou. Moreover, the interface of saltwater and fresh water declined at approximately 10 m, with a maximum depth of 30 m, threatening the fresh water in the deep aquifer (Han and Han, 2006).

Meanwhile, the concerns for the environment risk was accumulating with the aquifer degradation. In 1997, the Irrigation and Drainage Group of Hebei Hydraulic Engineering Society organized a meeting on water saving agriculture, and pointed out the environmental problems related to the deterioration of the aquifer, and appealed humans to conduct high efficiency water saving. The Disaster Prevention and Mitigation Group of Hebei Hydraulic Engineering Society, which was established in 1996, committed themselves to change the idea of drought relief from well drilling to integrated measures. In 2002, they suggested the government and water conservancy bureau to take water saving and water conservation as a priority, and to reduce groundwater withdrawal. Above events revealed an obvious increase of the community sensitivity.

Corresponding measures were taken. In 1998, Hebei Government issued several provisions of water saving for the whole society. Subsequently, water-saving irrigation was identified as the focus of water conservancy constructions in Cangzhou in 1998. Investments in water-saving projects were enhanced, and subsidies were provided. Accordingly, the irrigated area with water-saving technologies (mainly low-pressure pipeline irrigations and sprinkler irrigations) in Cangzhou rapidly increased from 96.4 $km^2$ in 1995 to 212.5 $km^2$ in 2012. In 2002, the Hebei Provincial Government issued to further strengthen the groundwater resources management, and specified the over-exploited regions and the severely over-exploited regions in the plain area. The entirety of Cangzhou was included in the list of regions with severely over-exploited deep aquifer. With respect to the shallow aquifer, the over-exploited and severely over-exploited regions were 406 and 525 $km^2$ in area, jointly accounting for 6.6% of the total area of Cangzhou. As stipulated, in over-exploited regions, groundwater exploitation should be strictly controlled, and the expansion of the exploitation scale is prohibited.

### 3.5 Era 5 (2003–present): Returning the balance

During the drought, the groundwater crisis in the NCP once again gained widespread concern, and several measures were implemented at the national, provincial, and regional levels. In 2003, the Ministry of Water Resources issued the Suggestions Concerning the Strengthening of the Management of Groundwater Over-Exploitation Regions, in which several targets and measures for groundwater management were proposed. This is a landmark event starting Era 5. In 2004, a leading group headed by the executive vice mayor was established to prevent and manage the subsidence in Cangzhou. Measures for sustainable groundwater management was even emphasized in the Cangzhou Government Work Report in 2004. In 2005, the Hebei Provincial Government published a notice stating that groundwater exploitation should be restricted, and urban wells out of the management of the water resources department should be shut down. The Cangzhou Government subsequently began to shut down these wells in Cangzhou City. To adopt the restrictions on groundwater exploitation, investments in water-saving projects were enhanced, and subsidies were provided in Cangzhou. Accordingly, the irrigated area with water-saving technologies in Cangzhou rapidly increased to 3,526.5 $km^2$ in 2012, which accounts for 65% of the total irrigated area. At the same time, inter-basin water transfer project, brackish water, and other municipal projects were

also implemented. The water supply volume from non-groundwater sources (mainly the brackish water and the water transfer from other basins) significantly increased (Figure 2(b) and Table 3).

Table 3. Changes in natural variability and social drivers from Era 2 to Era 5

| | Annual precipitation (mm) | Surface inflow ($10^6$m$^3$) | Number of motor wells ($10^3$) [a] | Alternatives ($10^6$m$^3$) | | Water-saving irrigation Area ($10^3$km$^2$) [a] | Economic conditions | |
|---|---|---|---|---|---|---|---|---|
| | | | | Brackish water | Inter-basin transfer | | Cost | Subsidy |
| Era 2 | 544.1 | 2144 | 51.6 | 0 | 0 | 26.7[a] | Low | Well drilling |
| Era 3 | 554.9 | 578 | 93.6 | 0 | - | 96.42 | Middle | No Subsidy |
| Era 4 | 391.7 | 1258 | 95.0 | 19.0[b] | 41.2[c] | 212.48 | High | Water-saving |
| Era 5 | 547.2 | 1185 | 92.3 | 36.5 | 71.7 | 352.65 | High | Water-saving |

[a] The value at the end of each era; [b] The average value of 2001–2002, when it was used at large scale.

Remarkable effects were obtained from the implementation of these measures. The shallow groundwater withdrawal decreased from $360.4 \times 10^6$ m$^3$ in 2003 to $258.2 \times 10^6$ m$^3$ in 2013. Consequently, the shallow groundwater table rose again to 5.28 m beneath the ground in 2013, with an annual rise of 0.2 m. The deep groundwater withdrawal decreased from $1,011 \times 10^6$ m$^3$ in 2002 to $743.2 \times 10^6$ m$^3$ in 2013. The water table at the center of the depression cone of the deep aquifer rose to 73.47 m beneath the ground in 2013, with an annual rate of 3.0 m. According to the latest groundwater over-exploited regions specified by the Hebei Provincial Government in 2014, an area of 413 km$^2$ is alleviated from a seriously over-exploited region to an over-exploited region of the deep aquifer. An area of 525 km$^2$ is alleviated from a seriously over-exploited region to an over-exploited region of the shallow aquifer, and the former 406 km$^2$ over-exploited region in terms of shallow aquifer is abolished. Accordingly, the environment of the aquifers has been restored. Subsidence in Cangzhou City has been effectively controlled since 2005 (2005 Yearbook of Cangzhou). For example, in a monitoring site near Cangxian County, the subsidence rate decreased from 76 mm/a during 2001–2005 to 40 mm/a during 2007–2008 (Zhang et al., 2014). Nonetheless, the aquifers still suffer from serious environmental problems. The average water table of the aquifer is still deep although the depth in 2013 is equal to that in 1984, as the center of the depression cone moved from urban areas to the rural areas (Fig. 3). According to the Geological Environment Bulletin of Hebei of 2013, the depression cone of the deep aquifer in Cangzhou is 5,551 km$^2$.

In 2012, China launched the strictest water resource management strategy in which many strict groundwater management and protection measures were proposed. Subsequently, the Hebei Government and the Cangzhou Government published the implementation schemes. According to the implementation scheme, groundwater exploitation in Cangzhou will be strictly controlled. Except for the purpose of obtaining water for domestic use, the construction of new motor-pumped wells will not be approved. Groundwater exploitation is prohibited in urban regions already covered by a public water supply network. According to the High-Efficiency Water-Saving and Irrigation Program in Cangzhou launched in 2013, overall water savings in agricultural production will be achieved by 2020. Since the South-to-North Water Diversion Project was established in 2015, which annually diverts about $483 \times 10^6$ m$^3$ of water to Cangzhou, all wells in urban regions

not controlled by the water resources administrators will be shut down and groundwater over-exploitation in rural regions will be gradually cut down.

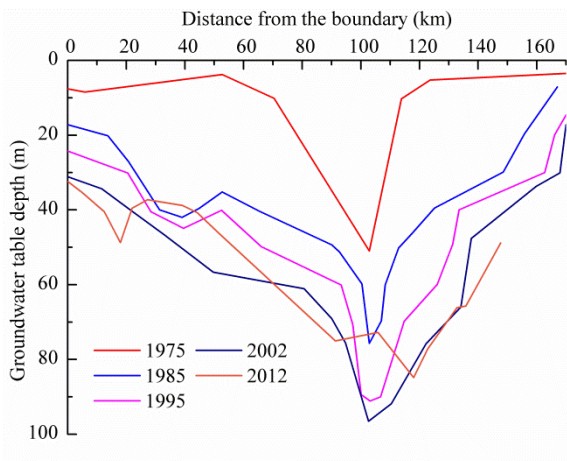

**Figure 3. Changes in the water table depth of the Aquifer Layer III along the cross-section from the west to the east.**

**4 Discussions**

**4.1 Interactions of the human–groundwater system**

According to the preceding analysis on the co-evolution of the human–groundwater system in Cangzhou, the specific application of the Taiji-Tire model for the human-groundwater system is provided in Fig. 4(a). The changes in natural variability (precipitation and surface inflow), and social productive force in Cangzhou through five eras are shown in Fig.

4(b). Human groundwater utilization is the main cause of the varying groundwater table. With increasing groundwater exploitation, both the shallow and deep groundwater table declined, and groundwater depression cone extended in Cangzhou. The contribution of increasing shallow groundwater utilization to the decline in the groundwater table is evident in the positive correlation between the depth fluctuations of the annual shallow water table (a negative change indicates a rise of the groundwater table) and the water withdrawal (Fig. 5(a)). The increasing deep groundwater utilization contributed to the

decline in the water table of the depression cone before 2002. The subsequent rapid reduction in deep groundwater utilization caused the water table at the center of the depression cone of deep groundwater to quickly rise again (Fig. 5(b)).

The groundwater sub-system also affects the behavior of water users. The significant decline in the shallow water table prevents the use of many shallow wells as before. By contrast, the water in the deep aquifer is guaranteed in terms of both quantity and quality albeit with higher cost. Consequently, groundwater utilization was directed toward the deep aquifer. In

1976, the average shallow water table depth was 2.59 m, and the water withdrawal from the deep aquifer was only 47.4% of that from the shallow aquifer. The ratio of deep water to shallow water withdrawal increased as the shallow water table declined from 1976 to 2002, especially during the drought in 1997–2002. In 2002, which had the deepest shallow water table, the water withdrawal from the deep aquifer was 2.82 times of that from the shallow aquifer. This feedback of the

groundwater sub-system to the human groundwater utilization can be detected from the high correlations between the ratio of deep to shallow water withdrawal and the shallow water table depth before 2002 (Fig. 5(c)).

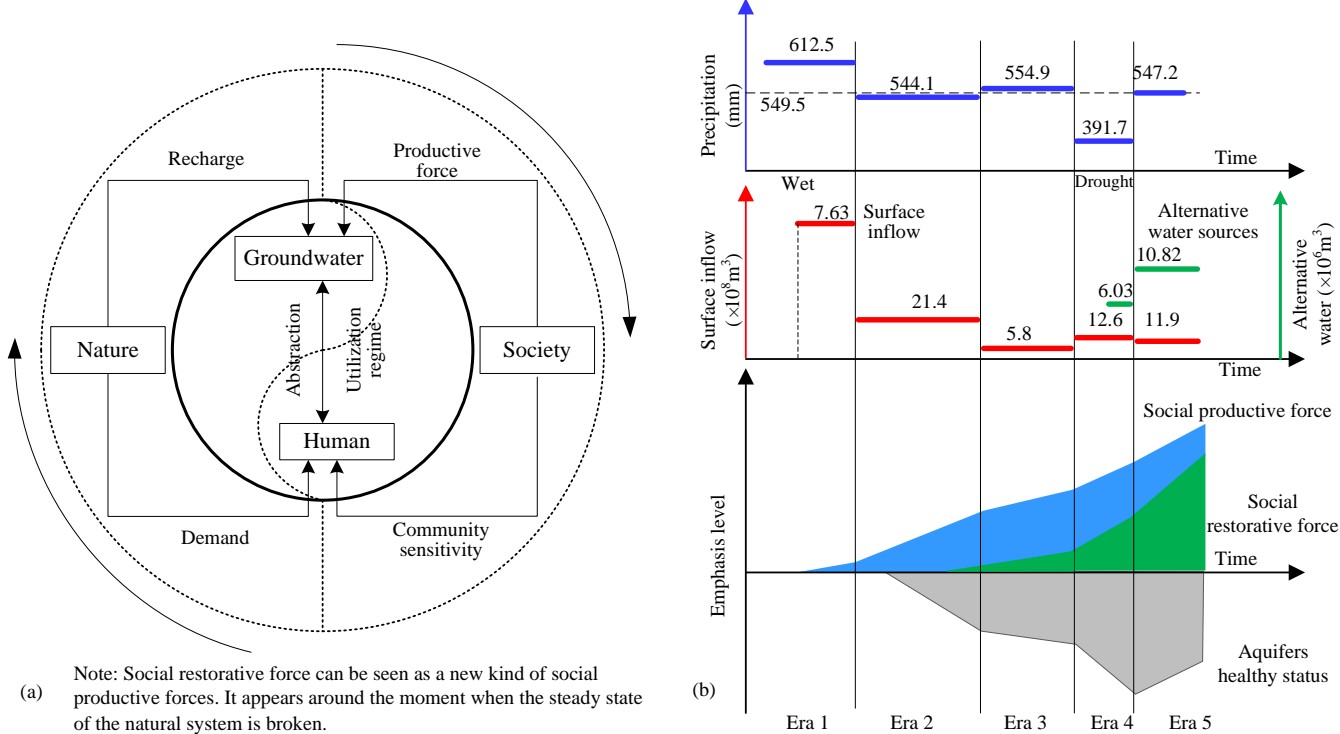

(a)

Note: Social restorative force can be seen as a new kind of social productive forces. It appears around the moment when the steady state of the natural system is broken.

(b)

5  **Figure 4. (a) Taiji-Tire model representation of the interactions of the human–groundwater system; (b) Changes in natural variability (precipitation and surface inflow), social productive forces in Cangzhou through five eras (Emphasis level used in relation to the vertical axis refers to the degree of increase in the variables described in the figure).**

The co-evolution of the human–groundwater system in Cangzhou is driven by natural variability, particularly the change/variability of surface water and precipitation. With decreasing surface water resources, the recharge to the
10  groundwater system decreased, and the demand for groundwater increased. Besides, the precipitation variability affected both the groundwater and human sub-systems. A larger precipitation implies a larger recharge to the shallow groundwater. As shown in Fig. 6(a), the changes in shallow water table depth are highly negatively correlated with the annual precipitation. The drought years with low precipitation are characterized with not only small recharge to the shallow aquifer but also a large demand for groundwater. Thus, groundwater withdrawal is negatively correlated with annual precipitation (Fig. 6(b)).

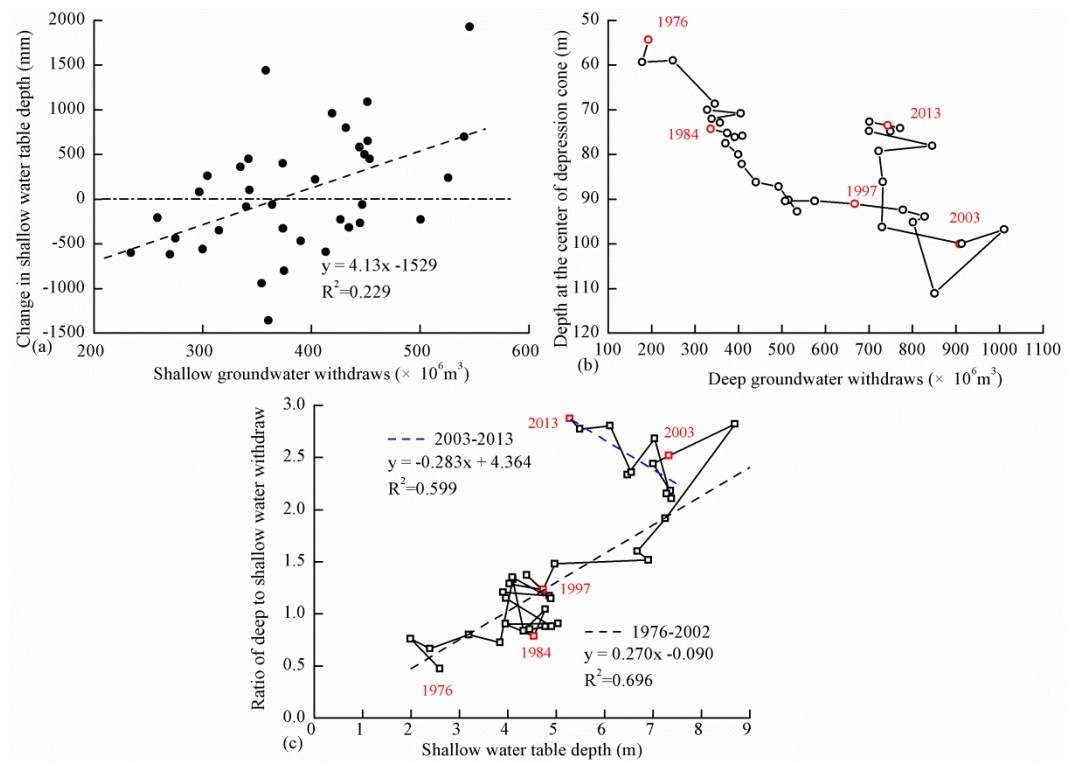

**Figure 5. (a) Relationship between shallow water table changes and withdrawal; (b) co-evolution of the depth of the depression cone center of deep aquifer with water withdrawal from 1976 to 2013; (c) the ratio of deep to shallow water withdrawal against the shallow water table depth before and after 2002.**

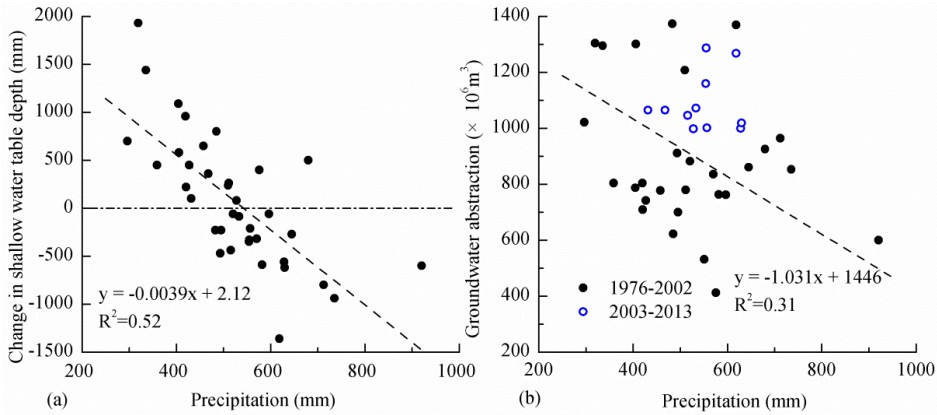

**Figure 6 (a) Change in shallow water table depth; (b) total water withdrawal against annual precipitation.**

The interactions of the inner Taiji usually lead to over-exploitation of natural resources and serious problems in ecosystems, which would be the main reason for the deterioration of the aquifers in Cangzhou. The feedback of the outer Tire will educate society of the negative effect of over-exploitation and therefore promote the development of new

10 responsive social behaviors to protect the environment and reduce the impulse of blind development. In Cangzhou, the

community sensitivity rose accompanied with the deterioration of the aquifers, represented by the event that Cangzhou Hydraulic Engineering Society submitted an appeal on water resources utilization to the government. Subsequently, environmental protection actions were taken, and they can be considered a new kind of social productive forces, which can be called social restorative forces. The social restorative force is not the same as the natural variations and is not an opposite force to social productive forces, which can also be restorative. The only difference is the social norms and values. In other words, humans themselves must decide how to use technology and devise policies. The social productive force only emphasizes the production but not the cost, including the direct production cost and the environmental externalities, whereas the social restorative forces refer to the specific social productive forces that further increase the production by lowering the environmental externalities. Therefore, the social restorative force can be regarded as a sub-branch of social productive force (with social destructive force being the other sub-branch). The social restorative force appears around the moment when the steady state of the natural system is broken, and increases with increasing community sensitivity. When the social restorative force dominates the social productive force, social productivity can be seen as green productivity (Tuttle and Heap, 2007; Mohanty and Deshmukh, 1998); that is, the societies can be seen as green (the other type is technological) (Baldassarre et al., 2015).

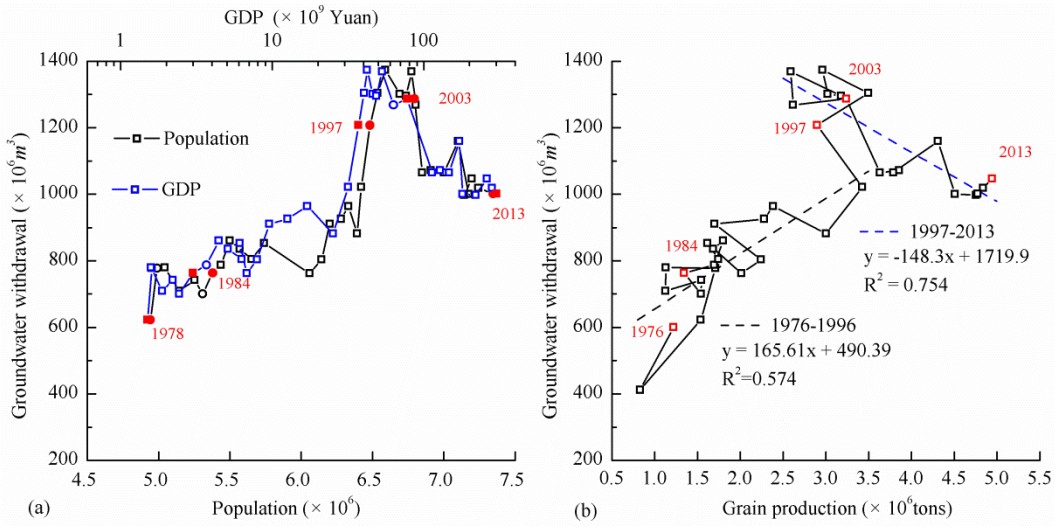

**Figure 7. (a) Relation between annual groundwater withdrawal and the population and GDP; (b) Co-evolution of the total groundwater withdrawal against the annual grain production before and after 2002.**

In Cangzhou, the efforts of controlling groundwater exploitation and water saving can be regarded as the social restorative force, which continued to increase (Figure 4(b)). However, the social restorative force was not strong enough before 2002, while the social productive force was dominated by that utilizes groundwater. Until 2002, groundwater withdrawal increased with the population and GDP of Cangzhou (Fig. 7(a)). As a result, the groundwater table declined rapidly. However, broken stationary condition of natural systems suggests higher levels of uncertainty and risks of extreme hydrometeorological events, causing higher social awareness of risks and resultant actions. The drought during 1997–2002

reminded the public and policymakers about taking measures to modify the existing development pattern. Given that economic development has always been the primary target of the government and society of Cangzhou, the mitigation measures for reallocating water from the economic development to the environment, which have been implemented in Murrumbidgee River basin (Kandasamy et al., 2014), are not acceptable. However, new policies are established to restrict

groundwater exploitation. The development and application of water-saving technologies are strongly encouraged to protect the environment while ensuring economic development, even if such strategies are more costly than traditional groundwater exploitation. Since 2002, after decades of accumulation, the community sensitivity toward aquifer environment has caused intensive actions toward environmental protection. These new improvements can be seen as an effort of humans to raise the social restorative force. As shown in Fig. 2(f), the area of water-saving irrigation has increased rapidly since then, and the

social productive forces have therefore reached a critical point.

The pattern of the human–groundwater system would be significantly changed since the moment when the social restorative force dominates the productive force. For the inner Taiji, significant changes in water user behavior and human response to groundwater system can be observed if the dataset was divided into two periods (before and after 2002) based on the narrative of the different eras. The ratio of deep water to shallow water withdrawal was negatively correlated with the

shallow water table depth after 2002, which is absolutely different from that before (Fig. 5(c)). However, the correlation of the negative relationship is considerably weaker at a starting point before 2002 (for example, the coefficient of determination ($R^2$) is only 0.11 according to the data from 2002 to 2013). As shown in Figs. 2(g) and 2(h), the deep groundwater withdrawal began to decrease slowly since 2002, whereas the shallow groundwater withdrawal continued to decrease rapidly as before. This finding indicates that people did not turn to shallow water as the shallow water table increased, because the

infrastructure of deep water with high-quantity exploitation already existed.

Supported by theses social productive forces, the pattern of the out tire was also significantly changed. On the one hand, water demand would no longer subtly vary with precipitation. The groundwater withdrawal decreased continually after 2002, although the annual precipitation kept stable. The significant negative correlation between the groundwater withdrawal and annual precipitation before 2002 ($R^2$ is 0.31) (Fig. 6(b)) would decrease if the period is extended to after 2002 ($R^2$ is only

0.25 when the used data is during 1976–2003). This decoupling also reveals an increased ability to mitigate climate variability, although an uncertainty still exists because the precipitation did not vary significantly during 2003–2013. On the other hand, the economic development did not deeply rely on increasing groundwater utilization. For example, the grain production grew rapidly again with the subsequent decline in groundwater utilization after 2002 (Fig. 2(d)). Thus, the grain production was negatively correlated with the groundwater withdrawal during 1997–2013 (correlation coefficient of −0.87),

which is different from that during 1976–1996 (correlation coefficient of 0.77) (Fig. 7(b)).

### 4.2 Roles of droughts during the evolution of the human–groundwater system

As an external force, drought influences the human-groundwater systems both directly through replenishments by recharge and indirectly through groundwater demand by discharge (Taylor et al., 2013). Drought often appears as shock, and

can significantly alters the system configuration. However, the human feedback processes responding to the drought are often relatively slower. As a non-linear system, the pulse drought process interacts with these slow processes, and results in complex and rich dynamics (Sivapalan et al., 2012). The two droughts (occurred in 1965 and during 1997-2002) played important roles in the coevolution of the human–groundwater system in Cangzhou, and they were chosen as critical events of

Era 2 and Era 4. However, these two droughts resulted in different types of regime shift of the system.

When the first drought occurred in 1965, humans in Cangzhou responded to ad hoc drought by drilling emergency wells. As the "environmental capacity" of the aquifers were adequate, the community sensitivity was not stimulated obviously. Conversely, the benefits from the emergency well stimulated groundwater withdrawals. Groundwater exploitation begun to be driven by the need for food. Therefore, a regime shift occurred, which pushed the system from natural status to human

exploiting. As the system remained within critical thresholds, it can be regarded as threshold change (Gordon et al., 2008). When the second drought occurred during 1997-2002, the environmental problems of the aquifers had been already serious, and the community sensitivity had accumulated to a high level. The unregulated groundwater withdrawals from the emergency wells and existing wells resulted in a sharp deterioration of the aquifers. Meanwhile, the community sensitivity increased rapidly, and resulted in that the social productive force exceeded a tipping point when the social restorative force

dominates the productive force. As the restorative force was strong enough to stimulate the system back to the balance, this regime shift can be regarded as hysteresis (Gordon et al., 2008). However, there is another regime shift which should be avoided, that is irreversible change (Gordon et al., 2008). It would occur if the restorative force triggered by the drought is not strong enough to stimulate the system back to the balance. The ability of the system to restore will be lost during a collapse, such as the ruin of many well-known ancient human settlements along the Silk Road (Liu et al., 2014). Although

the measures of water saving and restrictions of groundwater exploitation in Cangzhou, which began to be accelerated during the drought, have helped the system back to the balance, the crises management for drought should be substituted with drought preparedness and risk management, and the drought management plans must contain planned use of groundwater sustainable in the long- term (Gurdak, 2012).

**5 Conclusions**

The historical socio-hydrological analysis in Cangzhou enabled the recognition of the "pendulum swing" of the co-evolution of the human–groundwater system. The Taiji-Tire model was used to interpret the interaction and co-evolutionary dynamics of the coupled human–groundwater system in Cangzhou. At the first era, the intensity of groundwater exploitation was low. The human–groundwater system was primarily dominated by natural factors. No records on groundwater crisis induced by human activities was found. At the second era, groundwater was first exploited to combat the drought. Thereafter,

groundwater exploitation was driven by the need for grain yield. Moreover, with the decrease in surface water resources, the intensity of groundwater exploitation was elevated, and the human–groundwater system was driven by the social productive force. Meanwhile, intensive human activities led to the deterioration of the aquifers environment, which drew considerable concerns from society. At the third era, comprehensive management measures were implemented to address the groundwater

crisis, and the deterioration was mitigated to a certain extent. At the fourth era, drought appeared as a shock, which terminated the mitigation measures. Consequently, the intensity of groundwater exploitation rapidly increased, leading to the dramatic deterioration of the aquifers environment. At the fifth era, the drought was eased, and several measures were implemented to reduce groundwater exploitation. Water-saving technologies became acceptable economically and ideologically, and the aquifers environment began to be restored. The strictest water resource management scheme was launched in 2012, and the South-to-North Water Diversion Project was commenced in 2015. Further restoration of the aquifers environment is anticipated.

The two droughts resulted in different types of regime shift of the human–groundwater system in Cangzhou. The first drought in 1965 induced the system from natural status to human exploiting, the regime shift of which can be regarded as threshold change, as the system remained within critical thresholds. During the second drought from 1997 to 2002, people responded to ad hoc drought by drilling emergency wells or relying on unregulated groundwater withdrawal from existing wells. Nevertheless, accompanied with the dramatic deterioration of the aquifers environment, community sensitivity concerning water crisis was enhanced, which triggered a tipping point when the social restorative force dominates the productive force. Then, the system is stimulated to the balance, the regime shift of which can be regarded as hysteresis. However, the drought remains an unexplored variable of the returning the balance. A comprehensive and coordinated drought management plan should be implemented to avoid the irreversible change of the system.

**Acknowledgments**

This research was partially sponsored by the National Key R&D Program of China(Grant No. 2016YFC0402707), theNational Natural Science Foundation of China (Grant No. 51579249), Ministry of Science and Technology (Grant No. SQ2016YFSF020004), the Research Fund (Grant No. ID0145B292016) of China Institute of Water Resources and Hydropower Research. We acknowledge valuable input and feedback from members of the Panta Rhei Working Group on Socio-hydrologic Modelling and Synthesis, and on Comparative Study on the Co-evolution of Coupled Human-Water Systems.

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
