# Peer review of "Socio-hydrologic perspectives of the co-evolution of humans and groundwater in Cangzhou, North China Plain"

_Hydrology and Earth System Sciences, 2016_

## Referee Comment (RC1) · Anonymous Referee #1 · 21 Sep 2016

Han et al. present a well-researched historical case study on the groundwater use in Cangzhou in the North China Plain. The authors make the important point that hydrological outcomes, such as groundwater declines, are influenced by both social and natural forces. The case study data is a useful addition to the field and presents a good opportunity to test ideas and models generated through socio-hydrological analysis in other locations. However, the manuscript has left me with unanswered questions in terms of the paper objectives and methods applied. I believe the following comments will help the authors address these concerns.

General Comments

1) The objective(s) of the paper did not come across clearly. The authors state that

one objective of the paper is to "analyze the co-evolution of the human-groundwater system in Cangzhou throughout history, focusing on the interactions between the social productive force and natural variability." Are there other objectives? The objective should also be introduced sooner to focus the reader. Both the "pendulum swing" concept and Taiji-Tire model are relatively new and untested ideas. The authors have not yet convinced me that they can assume these fit the case before completing the analysis. If assessing this fit is an additional paper objective, it should be made clear. Alternatively, if the paper seeks to further specify he model or test a specific aspect of it, that should also be clarified.

2) The methods section needs expanding as it is not clear what methods were used to develop and analyze the historical narrative. The authors use the concept of a "pendulum swing" to introduce and organize the narrative. However, it is not clear what criteria were used to determine if and when a "pendulum swing" occurs. Five eras are presented, what criteria were used to determine that a new era had begun? The Taiji-Tire model is used to frame the analysis. How was the case mapped to the Taiji-Tire? How, for example, was the spatial boundary of the internal Tire determined? And how were forces classified as productive or restorative?

3) The authors nicely demonstrate how variability in precipitation can alter the simple story of reaching a tipping point and adjusting behavior to adapt. I think this is a good contribution. However, in complex systems such as socio-hydrological systems there is great potential for multi-causality and teleconnections. In addition to groundwater levels and precipitation, were other drivers of water use behavior change considered? How were the historical narrative and data set used to focus on these drivers? Please clarify.

4) While the writing did not interfere with my ability to review the manuscript there are a substantial number of grammatical errors and instances of unclear syntax. I have pointed out several, but not all, of these below. Thorough proof reading is needed before publication.

Specific Comments

1) On Page 1, Line 30, define or explain what is meant by the term human forcing.

2) On Page 2, Line 2, define or explain what is meant by the term salience threshold.

3) In Section 2, the authors do a great job describing the hydrological and geological setting of the case. A paragraph on the governance and institutional structure of the study region would be an excellent addition here, particularly for international readers. This would help readers less familiar with Chinese governmental divisions better follow the roles of the various entities in the narrative.

4) On page 4, line 2, the authors specify the sources of hydrological, agricultural and water use data sources. However, it is not clear what the data source is for policy initiatives (Table 1) or how relevance of policy initiatives was determined.

5) On page 9, line 11, in the description of the drought era (1997-2002) the authors state that well drilling "seemed to be the only choice to resist the drought." Yet, in section 3.5 they describe measures such as water licensing (1999) and irrigation efficiency improvements (1998). Why aren't these measures discussed in conjunction with the expansion of well drilling?

6) On page 9, section 3.5, the description of era 5 (2003-present) contains several events that occur before 2003 such as the 1999 water licensing system. Why aren't these events considered as part of era 4?

7) On page 11, figure 3, I appreciate the qualitative plotting of the level of emphasis on production and restoration. However, I would like to understand how these levels were estimated. What data sources (either quantitative or qualitative) were used? I am also unsure of the meaning of the emphasis level of "healthy status" in this context. Does healthy refer to environmental or public health? And if it refers to environmental health how does the emphasis on environmental health differ from the focus on restoration (or the restorative force)? Please clarify.

8) Page 12, figure 4a and page 13 figure 5a: clarify the directionality of shallow water table changes. Is a negative change a decline in groundwater levels or a decrease in the depth the ground water table?

9) On page 12, figure 4c conveys change in the relationship between shallow groundwater table depth and the ratio of deep to shallow groundwater. The reader needs more information to properly interpret this figure. How was this data set separated into these two groups (before and after 2002)? Was the division determined solely based on the narrative or were statistical tests used? Does any of the qualitative historical data collected aid in interpretation of this plot? What does this plot illustrate about the behavior of water users in the basin?

10) On page 13, line 3, the authors emphasize that the social restorative force is not necessarily in opposition to the restorative force and can be considered a subset of the productive force. Is this a modification to the original Taiji-Tire model? Please clarify.

11) On page 13, lines 8 and 24 the authors make reference to the system steady state and the date in which it was broken. Please clarify what in this instance was in steady state as I am skeptical that the socio-hydrological system broadly defined was ever in steady state. Please also describe how 1965 was identified as the end to this steady state period.

12) On page 13, figure 5b, more information is also needed to interpret this figure including how the data set was divided into two periods.

13) On page 14, line 6 the authors discuss the accumulation of community sensitivity. What data, either quantitative or qualitative, can back up this statement? It would also be helpful to clarify what is meant by community sensitivity. In the article referenced, Elshafei et al (2014), the authors specify what community sensitivity is theorized to depend on, that would also be useful here.

14) On page 14, line 12 the authors state that costly new technologies are adopted
solely to protect the environment. Please note how were other motivations or causes ruled out.

15) On page 14, figure 6b same comment as figure 5b above.

16) On page 15, the authors note that groundwater withdrawal no longer varies with precipitation. What enabled this decoupling?

Technical Corrections

1) Page 1, Line28: Syntax is awkward: "Except for the social forcing, natural variability is another external forcing." Could rephrase as: "In addition to the social forcing, natural variability is an external forcing."

2) Page 2, Line 27: Correct grammatical errors: "Because that groundwater pumping from the aquifer increases obviously since middle of the 1960s, the NCP aquifer system becomes one of the most overexploited aquifer in the world" perhaps as: "Because groundwater pumping from the aquifer has increased significantly since middle of the 1960s, the NCP aquifer system has become one of the most overexploited aquifers in the world."

3) Page 9, line 2: missing closing parentheses after the word year.

4) Page 14, line 14: replace development with developed.

5) The citation Liu et al. (2014) is missing from the references.

---

## Referee Comment (RC2) · Anonymous Referee #2 · 11 Oct 2016

The authors present an interesting case on groundwater use in Cangzhou, North China Plain. The area is highly agricultural and co-evolution of the area has been dominated by the desire to keep agricultural produce high and mitigate unintended consequences resulting from water resource use. The paper, however, still has loose ends that need to be solved. Also a better alignment of objectives, methods and analysis, would improve readability. I hope the authors find the comments below constructive.

General comments

1) It would be helpful for the reader if concepts were clearly defined from the start. Currently, the use of concepts is mixed and includes the use of Taiji Tire model, the concept of pendulum swing (Kandasamy et al. 2014) and the concept of community

sensitivity (Elshafei et al., 2014). While the pendulum swing is not defined as such, the concept of community sensitivity is introduced in the discussion, but at the same time forms a major part of the discussion. If the authors wish to use more than one concept, the reader would benefit from a more comprehensive introduction of these concepts early in the paper, possibly including their purpose and/or limitations, and how these concepts are used for the current analysis of the Cangzhou case. The latter would give the reader a clearer indication of what it can and cannot expect from the current analysis. For example, the Taiji-Tire model is merely offered as an organizing framework to represent and explain the human-water relationships (Liu et al., 2014). The conceptualization of interactions serves as a first step to a quantitative (numerical) model that can be used to explain the past and develop predictive insights (Liu et al., 2014). The pendulum swing refers to "an exclusive focus on agricultural development and food production in the initial stages and its attendant socioeconomic benefits, followed by the gradual realization of the adverse environmental impacts, subsequent efforts to mitigate these with the use of remedial measures, and ultimately concerted efforts and externally imposed solutions to restore environmental health and ecosystem services" (Kandasamy et al., 2014).

2) The primary goal as defined by the authors is the interpretation of the case study using the Taiji Tire model. It remains however unclear which methods are used to relate the observed feedbacks in the case study to the more abstract concepts in the model: what is used to distinguish endogenous from exogenous variables? How are the major drivers of the system resolved? When is a feedback considered to be productive or restorative? Currently, his seems to be dependent on your system boundary? The environmental burden seems to be partly shifted from groundwater to the surface water systems that deliver the water transfers?

3) Environmental awareness/ community sensitivity/ natural restorative force is ultimately put forward as a driver of groundwater table restoration. It is however unclear at the moment how this is inferred from the case study. The description of the case study

is detailed and shows the complexity of the system including groundwater changes in both the shallow and deep water aquifer, resulting in policy developments aimed at improved groundwater management and ultimately an increase in groundwater tables. At the same time, it is mentioned that pumping costs increase due to the deeper groundwater table, land subsides (up to almost a meter), salt intrudes and additional water is available due to water transfers. How does environmental awareness relate to economic incentives or the availability of an alternative water source? If the paper would include an interpretation (qualitative or quantitative) of the strength of the various feedbacks, if would make the current conceptualization much stronger.

4) The text would benefit from editing and proof reading to improve its readability.

Specific comments

Page 3, line 3. Please lengthen, since it is currently the core of the article. By which measure are feedbacks categorized? How exactly is a social productive force defined? How are main drivers distinguished among all plausible drives? Zhang et al. (2011), for example use five steps to deduct causal mechanisms in their research to link climate change and large-scale human crises.

Page 4, line 12 to 16, please check era titles with paragraph titles; not all of them match.

Page 4, paragraph 3.1, what are the sources used for this paragraph?

Page 4, paragraph 3.1, while the statements related to irrigated area argue in favor of natural variability dominating socio-hydrological change, the statements related to reservoirs, diversion projects and drainage-oriented policy currently imply that humans seems to have considerable impact already during this period. Adding a statement on the (limited?) effects of these policies on groundwater would strengthen the argument.

Page 5, figure 2, please check the figure references: subfigure d is referred to twice, resulting in a mismatch (e to i) from there on.

Page 5, figure 2f (irrigated area), what is meant by irrigated area, is this total irrigated area or irrigated area using surface water?

Page 5, figure 2, having a figure here showing the incoming water from all or the most important diversions/transfers into the region would complete the story. I can imagine that the availability of an alternative water source plays a significant role in the restoration of groundwater levels.

Page 7, line 18, "Groundwater then became an important water resource for agricultural irrigation." Should this be "the most important water resource"? Given that well drilling for groundwater started a few years earlier?

Page 7 to 10, paragraphs 3.2, 3.3., 3.4, what are the sources used for these paragraphs?

Page 9, line 19. "the environment noticeably deteriorated". How was this the case?

Page 9 & 10, has the reduction of overexploitation become a goal in itself or are earlier mentioned problems such as subsidence or salt water intrusion still an issue in the region? Are there specific quotes from governmental documents you could use to strengthen your argument?

From Page 11 onwards, How, using what method/definitions, are the more abstract general concepts as mentioned in figure 3 related to the individual, observed feedbacks as mentioned in the text? How are productive, restorative and healthy status defined? Can anything be said about what triggered the restorative force (e.g. economic motives, a change of norms and values) and/or what is meant by the steady state of the system?

Page 12, figure 4a, how are the axis defined? Is a decrease in groundwater table indicated with a positive sign? Are values calculated with regard to the groundwater table of the previous year? Has a correction been applied for water inflow (e.g. precipitation)?

[Figure]

Page 12, figure 4b, out of interest, how can it be that with a larger withdrawal the center of depression in 2013 is equal to 1984?

Page 12, figure 4c. The description of this figure is quite difficult to follow. Since two sets of data are presented in the figure, i.e. black (1976 - 2002) and blue (2003 – 2013) a discussion of these different trends would be appreciated. If individual data points are discussed as is the case now, maybe the individual years could be marked in the plot?

Page 12, line 14. "The interactions of inner Taiji . . . of blind development." On which literature is this statement based? Elinor Ostrom has, among others, done a lot of research aimed at understanding the circumstances under which overexploitation takes place.

Page 13, line 13, it would be helpful to see the definition of the restorative and productive force earlier in the paper, for example when introducing the Taiji Tire model.

Page 14, line 6, "This is because at . . . protections (Elshafei et al., 2014)." On what evidence is this statement based? How is community sensitivity defined? How is it measured in the case study?

Page 14, line 13. "The social productive . . . extremely costly". On what are these statements based? Is there evidence that technology and management tools are developed solely for environmental protections? Costly in relation to what (alternative)?

Literature

Elshafei, Y., et al. "A prototype framework for models of socio-hydrology: identification of key feedback loops and parameterisation approach." Hydrology and Earth System Sciences 18.6 (2014): 2141-2166.

Kandasamy, J., et al. "Socio-hydrologic drivers of the pendulum swing between agricultural development and environmental health: a case study from Murrumbidgee River basin, Australia." Hydrology and Earth System Sciences 18.3 (2014): 1027-1041.

Liu, Ye, et al. "Socio-hydrologic perspectives of the co-evolution of humans and water in the Tarim River basin, Western China: the Taiji–Tire model." Hydrology and Earth System Sciences 18.4 (2014): 1289-1303.

Zhang, David D., et al. "The causality analysis of climate change and large-scale human crisis." Proceedings of the National Academy of Sciences 108.42 (2011): 17296-17301.

---

## Author Comment (AC1) · 4 Dec 2016

**Response to Referee #1**

General Comments

1) The objective(s) of the paper did not come across clearly. The authors state that one objective of the paper is to "analyze the co-evolution of the human-groundwater system in Cangzhou throughout history, focusing on the interactions between the social productive force and natural variability." Are there other objectives? The objective should also be introduced sooner to focus the reader. Both the "pendulum swing" concept and Taiji-Tire model are relatively new and untested ideas. The authors have not yet convinced me that they can assume these fit the case before completing the analysis. If assessing this fit is an additional paper objective, it should be made clear. Alternatively, if the paper seeks to further specify he model or test a specific aspect of it, that should also be clarified.

Response: There are two objectives of the paper, and we will state them clear in the revised manuscript: "The objectives of the paper are: (1) to chart the "pendulum swing" history of the groundwater utilization in Canagzhou, with a particular focus on the dynamics of human–groundwater interactions that resulted in the "pendulum swing" around the balance point in groundwater allocations between humans and aquifer ecosystems, as well as the natural variability and social forcings that contributed to it; (2) to interpret the interactions and co-evolutions of the human-groundwater system within Cangzhou using the Taiji-Tire model, with the incorporation of the concepts of restorative force and community sensitivity."

We agree with the comment. In the revised manuscript, we will add an introduction about the "pendulum swing" at first, and point out that the history of co-evolution of the human-groundwater system in Cangzhou fits a "pendulum swing":

"Linked by the processes of adaption, the human society co-evolves with the hydrological system, often resulting a "pendulum swing" around the balance point of the human-water system (Kandasamy et al. 2014, Sivapalan 2015). The concept of pendulum swing was first characterized by Kandasamy et al. (2014), through tracing the 100 years history of the competition for water between agricultural development and environmental health within the Murrumbidgee River Basin. Similar dynamics was also found in the human-water system in the arid Tarim Basin (Liu et al. 2014), and the case of human-flood interactions (Baldassarre et al. 2015). A pendulum swing can be specified into three or four typical stages: the initial exploitation stage exclusively focusing on economic development, the onset environmental degradation stage accompanying with the introduction of remedial infrastructure, the widespread environmental degradation stage led to mitigation measures, and the recovery stage along with the implementation of ultimately solutions.

We will add a more detailed introduction of the Taiji-Tire model, and clarify that the Taiji-Tire model is specified to the groundwater system, with the incorporation of the concepts of restorative force and community sensitivity.

"A specific social hydrological system contains human, hydrological and environmental sub-systems. The Taiji-Tire model proposed by Liu et al. (2014) is a framework to represent and explain a specific social-hydrological system under its outer environmental system. In the model, a Taiji wheel, a term from a special concept in Chinese philosophy, is used to describe the direct human–water relationship of a specific social hydrological system. While a human–water tire is used to represent the indirect impact of external natural and social factors

that affect the water. The pendulum swing of the water competition between agriculture and environment in the Tarim Basin in a long history was revealed by the Taiji-Tire model, with attention on the interactions between natural variability and social productive force (Liu et al. 2014). In order to interpret the drivers of the reallocation of water from social-economy to the environment under the recovery stage of the pendulum swing, a concept of "environmental restorative force", which is comparative to "social productive force" was proposed (van Emmerik et al. 2014). It should be noted that the environmental protection actions during the recovery stage are usually conducted upon a high social awareness of environment risk or welfare (Di Baldassarre et al. 2013). Elshafei et al. (2014) uses a new concept of community sensitivity to represent this social awareness of environment welfare. Community sensitivity is the sensitivity of human society to the changing environment. High community sensitivity represents that humans feel the pressure of environmental deterioration, and tend to restrain human activities to restore environmental health. The concept of community sensitivity was used to analyze the switching of the favoring between flood protection and wetlands in the Kissimmee River Basin, Florida (Chen et al. 2016). Above new concepts can be incorporated into the Taiji-Tire model to improve its explanatory power on a specific human-water system."

2) The methods section needs expanding as it is not clear what methods were used to develop and analyze the historical narrative. The authors use the concept of a "pendulum swing" to introduce and organize the narrative. However, it is not clear what criteria were used to determine if and when a "pendulum swing" occurs. Five eras are presented, what criteria were used to determine that a new era had begun? The Taiji-Tire model is used to frame the analysis. How was the case mapped to the Taiji-Tire? How, for example, was the spatial boundary of the internal Tire determined? And how were forces classified as productive or restorative?

    Response: At first, we will introduce the concept and definition of "pendulum swing" in the "Introduction" section, especially the different stages of it (please refer to the response to the first comment). Then, we will explain the criteria used to determine if and when a "pendulum swing" occurs, and the criteria that a new era had begun as follows:

    The time series of well numbers and irrigated area (including water-saving irrigated area) were analyzed to detect the changes in infrastructure, under the background of social development revealed by the changes in population, and grain production. The average water table depth of the shallow aquifer, and the water table depth of the depletion cone of the deep aquifer were used as the main indices of the environmental health status. The break points of the time-series of the water table depth and groundwater withdrawal were obtained according to the changes of the trends. The major policies and initiatives that facilitated developments, or resulted in a turning point in the groundwater water management were examined to confirm the break points. Then, the co-evolution of the human-groundwater system was classified into different eras, which are used to determine if and when a "pendulum swing" occurs.

    We will explain the Taiji-Tire model as follows:

    The Taiji-Tire model is firstly used as a framework to explain how complex socio-hydrological systems coevolved with direct or indirect interactions between factors

from both human and water sides. For the case in Cangzhou, the groundwater is apparently influenced by humans during past several decades and in turn the varying groundwater tables were also shaping the community sensitivity of humans on the environment issues and drives humans to self-regulate theirs behaviors by establishing new policies and developing new technologies like water-saving technology. The most valuable contribution of the Taiji-Tire model is to provide an overall perspective of a socio-hydrological system with explicit presentation of the interactions between multi-factors that generate rich dynamics. It is not yet an accurate math or physical model, but a framework for scholars to study and exchange ideas on a specific case.

As for the productive and restorative forces, as mentioned in the paper, for example Fig. 3a, the restorative forces is explained as a new kind of social productive forces. The social productive force refers to the combination of all factors that help humans to utilize the resources and create better material and spiritual products that makes life better and easier. While the social productive force itself only emphasize the production but not the cost, including the direct production cost and the environmental externalities, the restorative forces refers to the specific productive forces that aiming at further increase the production by mainly lowering the environmental externalities, or, in another word, the green productivities.

3) The authors nicely demonstrate how variability in precipitation can alter the simple story of reaching a tipping point and adjusting behavior to adapt. I think this is a good contribution. However, in complex systems such as socio-hydrological systems there is great potential for multi-causality and teleconnections. In addition to groundwater levels and precipitation, were other drivers of water use behavior change considered? How were the historical narrative and data set used to focus on these drivers? Please clarify.

Response: We agree that the socio-hydrological system is multi-causality and teleconnections. The drivers of water use behavior change include precipitation, groundwater level, other water resources (surface inflow, water transfer from other basins, and brackish water), Water-saving irrigation area, and economic conditions (the cost of groundwater abstraction and the subsidy policy). In the revised manuscript, we will aggregate these drivers from era 2 to era 5 in a new table, and give a more detailed and concentrated historical narrative on theses drivers.

**Table 3. Changes in factors related to the community sensitivity from era 2 to era 5**

| | Precipitation mm | Surface inflow $10^6 m^3$ | Alternatives ($10^6 m^3$) | | Water-saving irrigation area ($10^3 km^2$) [a] | Economic conditions | |
| --- | --- | --- | --- | --- | --- | --- | --- |
| | | | Brackish water | Inter-basin transfer | | Cost | Subsidy objective |
| Era 2 | 544.1 | 2144 | 0 | 0 | 26.7[a] | Low | Well drilling |
| Era 3 | 554.9 | 578 | 0 | 0 | 96.42 | Middle | No Subsidy |
| Era 4 | 391.7 | 1258 | 38.1 [b] | 41.2[c] | 212.48 | High | Water-saving |
| Era 5 | 547.2 | 1185 | 36.5 | 71.7 | 352.65 | High | Water-saving |

[a] The value at the end of each era; [b] The value of 2002, when brackish water was used at large scale; [c] The average value of 2001 and 2002;

4) While the writing did not interfere with my ability to review the manuscript there are a

substantial number of grammatical errors and instances of unclear syntax. I have pointed out several, but not all, of these below. Thorough proof reading is needed before publication.

Response: We will follow closed the suggestions made by referees, and improve the use of English. The revised manuscript will be edited by a native English speaker.

Specific Comments

1) On Page 1, Line 30, define or explain what is meant by the term human forcing.

Response: We has revised this term as "social forcing (such as increase in water demand for agricultural development)"

2) On Page 2, Line 2, define or explain what is meant by the term salience threshold.

Response: "salience threshold " should be "resilience thresholds" according to (Sivapalan et al. 2012)

3) In Section 2, the authors do a great job describing the hydrological and geological setting of the case. A paragraph on the governance and institutional structure of the study region would be an excellent addition here, particularly for international readers. This would help readers less familiar with Chinese governmental divisions better follow the roles of the various entities in the narrative.

Response: We agree and will add an instruction on the governance and institutional structure of the study region:

Cangzhou is a prefecture-level city of Hubei Province, and has 4 county-level cities and 10 counties. Cangzhou government abides by the provincial and national policies of water resources management, and makes policies for the whole region. The Water Resources Bureau of Cangzhou, as a department of Cangzhou Government, takes charge of the water resources affairs, and guides the Water Resources Bureau of the 10 counties and 4 county-level cities, and is also guided by the Water Resources Department of Hebei, the Ministry of Water Resources, and the Haihe River Water Resources Commission.

4) On page 4, line 2, the authors specify the sources of hydrological, agricultural and water use data sources. However, it is not clear what the data source is for policy initiatives (Table 1) or how relevance of policy initiatives was determined.

Response: The data of policies and initiatives before 1985 is discovered from the Water Resource Annals of Cangzhou (Xue 1994), the data after 1985 is detected from the announcements, documents of the Ministry of Water Resources, the Government of Hebei, the Water Resources Department of Hebei, the Government of Cangzhou, the Water Resources Bureau of Cangzhou. The break points of the time-series of the water table depth and groundwater withdrawal were obtained according to the changes of the trends. The major policies and initiatives that facilitated developments, or resulted in a turning point in the groundwater water management were examined to confirm the break points.

5) On page 9, line 11, in the description of the drought era (1997-2002) the authors state that well drilling "seemed to be the only choice to resist the drought." Yet, in section 3.5 they describe measures such as water licensing (1999) and irrigation efficiency improvements

(1998). Why aren't these measures discussed in conjunction with the expansion of well drilling?

Response: We will revise this sentence as "seemed to be the most immediate choice to resist the drought". The measures of water licensing (1999) and irrigation efficiency improvements (1998) will be discussed in conjunction with the expansion of well drilling.

6) On page 9, section 3.5, the description of era 5 (2003-present) contains several events that occur before 2003 such as the 1999 water licensing system. Why aren't these events considered as part of era 4?

Response: In the revised manuscript, the events that occur before 2003 (the 1999 water licensing system, irrigation efficiency improvements) will be moved to era 4.

7) On page 11, figure 3, I appreciate the qualitative plotting of the level of emphasis on production and restoration. However, I would like to understand how these levels were estimated. What data sources (either quantitative or qualitative) were used? I am also unsure of the meaning of the emphasis level of "healthy status" in this context. Does healthy refer to environmental or public health? And if it refers to environmental health how does the emphasis on environmental health differ from the focus on restoration (or the restorative force)? Please clarify.

Response: The emphasis level of the social productive force, can be detected from the changes in well numbers, the irrigated area with groundwater, as well as the policy for groundwater exploitation. While, the social restorative force can be detected from the changes in water saving irrigation area, as well as the policy to incent water-saving technologies.

"healthy status" is specified as "aquifers healthy status" in this context. The changes in the healthy status of the aquifers (both shallow and deep) can be detected from the changes of the average water table depth of the shallow aquifer, and the water table depth of the depletion cone of the deep aquifer. The changes of the emphasis on the aquifers healthy status are the consequences of the restorative forces.

8) Page 12, figure 4a and page 13 figure 5a: clarify the directionality of shallow water table changes. Is a negative change a decline in groundwater levels or a decrease in the depth the ground water table?

Response: We are sorry for the misleading. In the revised manuscript, we have revised it as "change in shallow water table depth", and added a explanation "a negative change means a rise of the groundwater table".

9) On page 12, figure 4c conveys change in the relationship between shallow groundwater table depth and the ratio of deep to shallow groundwater. The reader needs more information to properly interpret this figure. How was this data set separated into these two groups (before and after 2002)? Was the division determined solely based on the narrative or were statistical tests used? Does any of the qualitative historical data collected aid in interpretation of this plot? What does this plot illustrate about the behavior of water users in the basin?

Response: The dataset was divided into two periods (before and after 2002) based on the narrative of different eras, and significant changes in water user behavior and social response

to groundwater system could be found. The ratio of deep to shallow water withdrawal negatively correlated with the shallow water table depth after 2002, which is absolutely different from that before (Figure 4(c)). However, the correlation of the negative relationship is much weak with a start point before 2002 (for example, the determine coefficient is only 0.11 with data from 2002 to 2013). As shown in Fig. 2(h), the deep groundwater withdrawal began to decrease slowly since 2002, while the shallow groundwater withdrawal continued to decrease rapidly as before. It indicates that people did not turn to shallow water with increasing shallow water table since the infrastructure of deep water with high quantity exploitation has already existed.

10) On page 13, line 3, the authors emphasize that the social restorative force is not necessarily in opposition to the restorative force and can be considered a subset of the productive force. Is this a modification to the original Taiji-Tire model? Please clarify.

     Response: First of all, please refer to our response to the 2nd comment of 1st referee. Secondly, it is not a modification but a supplement to the original Taiji-Tire model. Liu et al. (2014) firstly addressed the idea of Taiji-Tire model. In their work, the socio-hydrological system are seen as consequence of interactions of two general drivers, namely natural variabilities and the social productive forces(Liu et al.., 2014). When Kandassamy et al. (2014) use this Taiji-Tire idea to their work in Murrubidgee Basin, they defined a new term of natural restorative force to substitute the natural variability as the opposite factor of social productive forces. In this study, we believe that this idea of natural restorative force is confusing and should be seen as a subtype of social productive forces.

11) On page 13, lines 8 and 24 the authors make reference to the system steady state and the date in which it was broken. Please clarify what in this instance was in steady state as I am skeptical that the socio-hydrological system broadly defined was ever in steady state. Please also describe how 1965 was identified as the end to this steady state period.

     Response: We focus on the human-groundwater system within Cangzhou, and the surface water is taken as an external driver. We means that the human-groundwater system within Cangzhou was in steady state before 1965, not the socio-hydrological system. We will explained why it is defined as a steady state and how 1965 was identified as the end to this steady state period in the revised manuscript.

     Before 1965, owing to the technological limitation, groundwater utilization was not be large scale. The volume of groundwater withdrawal was small relative to surface water, and irrigated area with groundwater was constrained to a small fraction of Cangzhou. As per the Taiji–Tire model, the human–groundwater relationship was weak, without any kind of sophisticated interactions. The human sub-system was not sensitive to the groundwater sub-system, and the groundwater sub-system was not affected by humans at a large scale. Therefore, the human-groundwater system can be considered stationary (or considered as in a steady state) without significant external drivers. In order to combat the drought in 1965, Cangzhou government proposed an expansive policy of groundwater utilization for the first time in history, and well drilling was rapidly accelerated. The relationship between the human and groundwater sub-systems has been enhanced since then, and the stationary condition was broken.

12) On page 13, figure 5b, more information is also needed to interpret this figure including how the data set was divided into two periods.

Response: We will add some information including how the data set was divided into two periods as follow:

Meanwhile, the groundwater withdrawal decreased continually after 2002, although the annual precipitation kept stable. The groundwater withdrawal was significantly negatively correlated with annual precipitation before 2002, with the coefficient of determination ($R^2$) 0.31. But the correlation significantly decreases if the period extends to the year after 2002 (the $R^2$ is only 0.25 using data of 1976-2003). This decoupling reveals that water demand would no longer subtly vary with precipitation, with the extension of water-saving technologies and the construction of water-saving projects, as well as the restricted groundwater exploitation policies. Although there is an uncertainty as the precipitation did not vary significantly during 2003-2013, it indicates an increase in the ability to mitigate the climate variability.

13) On page 14, line 6 the authors discuss the accumulation of community sensitivity. What data, either quantitative or qualitative, can back up this statement? It would also be helpful to clarify what is meant by community sensitivity. In the article referenced, Elshafei et al (2014), the authors specify what community sensitivity is theorized to depend on, that would also be useful here.

Response: We will add an introduction on community sensitivity at first:

It should be noted that the environmental protection actions during the recovery stage are usually conducted upon a high social awareness of environment risk or welfare, which was taken as a variable of the model (Di Baldassarre et al. 2013). Elshafei et al. (2014) uses a new concept of community sensitivity to represent this social awareness of environment welfare. Community sensitivity is the sensitivity of human society to the changing environment. The high community sensitivity represents that humans feel the pressure of environmental deterioration, and tend to restrain human activities to restore environmental health. The concept of community sensitivity was used to analyze the switching of the favoring between flood protection and wetlands in the Kissimmee River Basin, Florida (Chen et al. 2016).

14) On page 14, line 12 the authors state that costly new technologies are adopted solely to protect the environment. Please note how were other motivations or causes ruled out.

Response: We will add a statement as follow:

As economic development is always a primary target of the government and the society of Cangzhou, the mitigation measures of reallocating water from the economic development to the environment, which have been implemented in Murrumbidgee River basin (Kandasamy et al. 2014), is not acceptable. Therefore, new policies are established to restrict the groundwater exploitation. and improving and applying the water-saving technology are strongly encouraged aiming at protecting the environment on the premise of ensuring economic development after the drought, even that they are extremely costly compared to traditional groundwater exploitation. The social productive forces therefore reach a critical point.

15) On page 14, figure 6b same comment as figure 5b above.

Response: We will change the division into 1976-1996 and 1997-2013 in the revised manuscript, and add some information as follow:

Supported by theses productive forces, the grain production declined slightly during the drought 1997-2002, but grew rapidly again with decreasing groundwater utilization after 2002 (Fig. 2(b)). As a result, the grain production negatively correlated with the groundwater withdrawal during 1997-2013 (the correlation coefficient is -0.87), which is absolutely different from that during 1976-1996 (the correlation coefficient is 0.77) (Fig. 6(b)).

[Figure]

16) On page 15, the authors note that groundwater withdrawal no longer varies with precipitation. What enabled this decoupling?

Response: The extension of water-saving technologies and the construction of water-saving projects, as well as the restricted groundwater exploitation policies enabled this decoupling. Please also refer to the response to comment 12).

Technical Corrections
1) Page 1, Line28: Syntax is awkward: "Except for the social forcing, natural variability is another external forcing." Could rephrase as: "In addition to the social forcing, natural variability is an external forcing."

Response: We will revise it following the suggestion.

2) Page 2, Line 27: Correct grammatical errors: "Because that groundwater pumping from the aquifer increases obviously since middle of the 1960s, the NCP aquifer system becomes one of the most overexploited aquifer in the world" perhaps as: "Because groundwater pumping from the aquifer has increased significantly since middle of the 1960s, the NCP aquifer system has become one of the most overexploited aquifers in the world."

Response: We will revise it following the suggestion.

3) Page 9, line 2: missing closing parentheses after the word year.

Response: We will add the closing parentheses.

4) Page 14, line 14: replace development with developed.
   Response: We will revise it following the suggestion.

5) The citation Liu et al. (2014) is missing from the references.
   Response: We will add the reference.

Reference:

Kandasamy, J., Sounthararajah, D., Sivabalan, P., Chanan, A., Vigneswaran, S. and Sivapalan, M. (2014) Socio-hydrologic drivers of the pendulum swing between agricultural development and environmental health: a case study from Murrumbidgee River basin, Australia. Hydrology And Earth System Sciences 18(3), 1027-1041.

Sivapalan, M. (2015) Debates-Perspectives on socio-hydrology: Changing water systems and the "tyranny of small problems"-Socio-hydrology. Water Resources Research 51(6), 4795-4805.

Liu, Y., Tian, F., Hu, H. and Sivapalan, M. (2014) Socio-hydrologic perspectives of the co-evolution of humans and water in the Tarim River basin, Western China: the Taiji–Tire model. Hydrology And Earth System Sciences 18(4), 1289-1303.

Baldassarre, G.D., Viglione, A., Carr, G., Kuil, L., Yan, K., Brandimarte, L. and Blöschl, G. (2015) Debates—Perspectives on socio-hydrology: Capturing feedbacks between physical and social processes. Water Resources Research 51(6), 4770-4781.

van Emmerik, T.H.M., Li, Z., Sivapalan, M., Pande, S., Kandasamy, J., Savenije, H.H.G., Chanan, A. and Vigneswaran, S. (2014) Socio-hydrologic modeling to understand and mediate the competition for water between agriculture development and environmental health: Murrumbidgee River basin, Australia. Hydrology And Earth System Sciences 18(10), 4239-4259.

Di Baldassarre, G., Viglione, A., Carr, G., Kuil, L., Salinas, J.L. and Blöschl, G. (2013) Socio-hydrology: conceptualising human-flood interactions. Hydrology And Earth System Sciences 17(8), 3295-3303.

Elshafei, Y., Sivapalan, M., Tonts, M. and Hipsey, M.R. (2014) A prototype framework for models of socio-hydrology: identification of key feedback loops and parameterisation approach. Hydrology And Earth System Sciences 18(6), 2141-2166.

Chen, X., Wang, D., Tian, F. and Sivapalan, M. (2016) From channelization to restoration: Sociohydrologic modeling with changing community preferences in the Kissimmee River Basin, Florida. Water Resources Research 52(2), 1227-1244.

Sivapalan, M., Savenije, H.H.G. and Blöschl, G. (2012) Socio-hydrology: A new science of people and water. Hydrological Processes 26(8), 1270-1276.

Xue, G. (1994) Water resouces annals of Cangzhou (in Chinese), Science and technology literature press, Beijing.

---

## Author Comment (AC2) · 4 Dec 2016

**Response to the second reviewer**

General comments

1) It would be helpful for the reader if concepts were clearly defined from the start. Currently, the use of concepts is mixed and includes the use of Taiji Tire model, the concept of pendulum swing (Kandasamy et al. 2014) and the concept of community sensitivity (Elshafei et al., 2014). While the pendulum swing is not defined as such, the concept of community sensitivity is introduced in the discussion, but at the same time forms a major part of the discussion. If the authors wish to use more than one concept, the reader would benefit from a more comprehensive introduction of these concepts early in the paper, possibly including their purpose and/or limitations, and how these concepts are used for the current analysis of the Cangzhou case. The latter would give the reader a clearer indication of what it can and cannot expect from the current analysis. For example, the Taiji-Tire model is merely offered as an organizing framework to represent and explain the human-water relationships (Liu et al., 2014). The conceptualization of interactions serves as a first step to a quantitative (numerical) model that can be used to explain the past and develop predictive insights (Liu et al., 2014). The pendulum swing refers to "an exclusive focus on agricultural development and food production in the initial stages and its attendant socioeconomic benefits, followed by the gradual realization of the adverse environmental impacts, subsequent efforts to mitigate these with the use of remedial measures, and ultimately concerted efforts and externally imposed solutions to restore environmental health and ecosystem services" (Kandasamy et al., 2014).

Response: We agree with the comment. In the revised manuscript, we will add an introduction about the "pendulum swing" at first, and point out that the history of co-evolution of the human-groundwater system in Cangzhou fits a "pendulum swing":

"Linked by the processes of adaption, the human society co-evolves with the hydrological system, often resulting a "pendulum swing" around the balance point of the human-water system (Kandasamy et al. 2014, Sivapalan 2015). The concept of pendulum swing was first characterized by Kandasamy et al. (2014), through tracing the 100 years history of the competition for water between agricultural development and environmental health within the Murrumbidgee River Basin. Similar dynamics was also found in the human-water system in the arid Tarim Basin (Liu et al. 2014), and the case of human-flood interactions (Baldassarre et al. 2015). A pendulum swing can be specified into three or four typical stages: the initial exploitation stage exclusively focusing on economic development, the onset environmental degradation stage accompanying with the introduction of remedial infrastructure, the widespread environmental degradation stage led to mitigation measures, and the recovery stage along with the implementation of ultimately solutions.

We also add a more detailed introduction of the Taiji-Tire model, and clarify that the Taiji-Tire model is specified to the groundwater system:

"A specific social hydrological system contains human, hydrological and environmental sub-systems. The Taiji-Tire model proposed by Liu et al. (2014) is a framework to represent and explain a specific social-hydrological system under its outer environmental system. In the model, a Taiji wheel, a term from a special concept in Chinese philosophy, is used to describe the direct human–water relationship of a specific social hydrological system. While a human–water tire is used to represent the indirect impact of external natural and social factors

that affect the water. The pendulum swing of the water competition between agriculture and environment in the Tarim Basin in a long history was revealed by the Taiji-Tire model, with attention on the interactions between natural variability and social productive force (Liu et al. 2014).

We will add an introduction of the concepts of restorative force and community sensitivity, and point out that they will be incorporated into the Taiji-Tire model:

In order to interpret the drivers of the reallocation of water from social-economy to the environment under the recovery stage of the pendulum swing, a concept of "environmental restorative force", which is comparative to "social productive force" was proposed (van Emmerik et al. 2014). It should be noted that the environmental protection actions during the recovery stage are usually conducted upon a high social awareness of environment risk or welfare, which was taken as a variable of the model (Di Baldassarre et al. 2013). Elshafei et al. (2014) uses a new concept of community sensitivity to represent this social awareness of environment welfare. Community sensitivity is the sensitivity of human society to the changing environment. The high community sensitivity represents that humans feel the pressure of environmental deterioration, and tend to restrain human activities to restore environmental health. The concept of community sensitivity was used to analyze the switching of the favoring between flood protection and wetlands in the Kissimmee River Basin, Florida (Chen et al. 2016). Above new concepts can be incorporated into the Taiji-Tire model to improve its explanatory power on a specific human-water system."

2) The primary goal as defined by the authors is the interpretation of the case study using the Taiji Tire model. It remains however unclear which methods are used to relate the observed feedbacks in the case study to the more abstract concepts in the model: what is used to distinguish endogenous from exogenous variables? How are the major drivers of the system resolved? When is a feedback considered to be productive or restorative? Currently, his seems to be dependent on your system boundary? The environmental burden seems to be partly shifted from groundwater to the surface water systems that deliver the water transfers?

Response: The inner Taiji represents the direct interacting human activities and hydrologic variables at short-term, reflecting the human-water relation. In Cangzhou, the Taiji represents the direct interactions between groundwater utilization and the water table. Therefore, he groundwater withdrawal from the shallow and deep aquifers, as well as the status of the aquifers (denoted by the shallow water table depth and water table depth of the depletion cone) are taken as the endogenous variables. The outer Tire represents all those social and natural factors that indirectly influence the system. Environmental change, especially precipitation and surface water change/variability, is an external driver of the human-groundwater system within Cangzhou. The social productive force is another external driver of the co-evolution of the human-groundwater system in Cangzhou. The social productive force is another external driver of the co-evolution of the human-groundwater system in Cangzhou. It should be noted that the Taiji-Tire model is specified to the groundwater system in Cangzhou. Therefore, the surface water systems is taken as a external driver.

As for the productive and restorative forces, as mentioned in the paper, for example Fig. 3a, the restorative forces is explained as a new kind of social productive forces. The social

productive force refers to the combination of all factors that help humans to utilize the resources and create better material and spiritual products that makes life better and easier. While the social productive force itself only emphasize the production but not the cost, including the direct production cost and the environmental externalities, the restorative forces refers to the specific productive forces that aiming at further increase the production by mainly lowering the environmental externalities, or, in another word, the green productivities. For Cangzhou, the emphasis level of the social productive force, can be detected from the changes in well numbers, the irrigated area with groundwater, as well as the policy for groundwater exploitation. While, the social restorative force can be detected from the changes in water saving irrigation area, as well as the policy to incent water-saving technologies.

3) Environmental awareness/ community sensitivity/ natural restorative force is ultimately put forward as a driver of groundwater table restoration. It is however unclear at the moment how this is inferred from the case study. The description of the case study is detailed and shows the complexity of the system including groundwater changes in both the shallow and deep water aquifer, resulting in policy developments aimed at improved groundwater management and ultimately an increase in groundwater tables. At the same time, it is mentioned that pumping costs increase due to the deeper groundwater table, land subsides (up to almost a meter), salt intrudes and additional water is available due to water transfers. How does environmental awareness relate to economic incentives or the availability of an alternative water source? If the paper would include an interpretation (qualitative or quantitative) of the strength of the various feedbacks, if would make the current conceptualization much stronger.

    Response: Environmental awareness is not only related to the measurable economic costs or incentives or the availability of an alternative water source. Community sensitivity is the sensitivity of human society to the changing environment (Elshafei et al. 2014). High community sensitivity represents that humans feel the pressure of environmental deterioration, and tend to restrain human activities to restore environmental health. The increasing economic costs would result in the increasing community sensitivity. The economic incentives can be taken as a response or adaption of the increasing community sensitivity. The water transfers may reduce the community sensitivity at a local scale during a short period, but the impacts can not last for long-term if the human-groundwater relationship is not change intrinsically.

4) The text would benefit from editing and proof reading to improve its readability.
    Response: We will follow closed the suggestions made by referees, and improve the use of English. The revised manuscript will be edited by a native English speaker.

Specific comments
Page 3, line 3. Please lengthen, since it is currently the core of the article. By which measure are feedbacks categorized? How exactly is a social productive force defined? How are main drivers distinguished among all plausible drives? Zhang et al. (2011), for example use five steps to deduct causal mechanisms in their research to link climate change and large-scale human crises.
    Response: We will make a detailed statement on the method about categorizing the

feedbacks, defining a social productive force and distinguishing plausible drives. Please refers to the response to 2nd comment of 2# referee. And also, we agree that distinguishing the main drivers from all other plausible drivers are important. For current, however, it still need more case studies to complete. We will try to answer this question in further works.

Page 4, line 12 to 16, please check era titles with paragraph titles; not all of them match.
    Response: We will check the era titles.

Page 4, paragraph 3.1, what are the sources used for this paragraph?
    Response: The data is from the Water Resource Annals of Cangzhou (Xue 1994). We will add the citation.

Page 4, paragraph 3.1, while the statements related to irrigated area argue in favor of natural variability dominating socio-hydrological change, the statements related to reservoirs, diversion projects and drainage-oriented policy currently imply that humans seems to have considerable impact already during this period. Adding a statement on the (limited?) effects of these policies on groundwater would strengthen the argument.
    Response: Since we focus on human-groundwater system, so the title will be revised to "Natural variability dominates human-groundwater system". In Cangzhou, "reservoirs, diversion projects" were conducted on surface water resources, and the "drainage-oriented policy" was restrained to low lands. The groundwater sub-system was not affected by humans at a large scale. We will add a statement as follow:
    Because of serious salinization problems, a drainage-oriented policy for low lands was established in Cangzhou. A large number of reservoirs and diversion projects were constructed, which reduced the need for groundwater resources.
    Therefore, owing to the low demand and the technological limitation, the scale of groundwater utilization was very small during this era. The relationship between the human and groundwater was weak. The society was more sensitive to the natural variability than the groundwater change, and the groundwater sub-system was not affected by humans at a large scale.

Page 5, figure 2, please check the figure references: subfigure d is referred to twice, resulting in a mismatch (e to i) from there on.
    Response: We will revise the figure references.

Page 5, figure 2f (irrigated area), what is meant by irrigated area, is this total irrigated area or irrigated area using surface water?
    Response: It means total irrigated area. We have revised it.

Page 5, figure 2, having a figure here showing the incoming water from all or the most important diversions/transfers into the region would complete the story. I can imagine that the availability of an alternative water source plays a significant role in the restoration of groundwater levels.
    Response: We have added a figure for the variations of water diversion from outside the

basin and the amount of brackish water. We think the new figure can be help.

[Figure]

Page 7, line 18, "Groundwater then became an important water resource for agricultural irrigation." Should this be "the most important water resource"? Given that well drilling for groundwater started a few years earlier?

Response: During this era, surface water was still more important than the groundwater. In 1983, the irrigated area with groundwater was 43.1% of the total irrigated area of Cangzhou. We have revised it as "Groundwater then became more and more important as a source for agricultural irrigation."

Page 7 to 10, paragraphs 3.2, 3.3., 3.4, what are the sources used for these paragraphs?

Response: The data used for paragraph 3.2 is from the Water Resource Annals of Cangzhou (Xue 1994). The data of agricultural infrastructures and production used for paragraphs 3.3, 3.4, and 3.5 is from the National Bureau of Statistics of China (2010), the Hebei Rural Statistics yearbook (from 1994 to 2013). The data of annual precipitation, groundwater withdrawal from both the shallow and deep aquifers, groundwater table depth is from the Hydrology and Water Resources Investigation Bureau of Cangzhou. The data of policies and initiatives before 1985 is acquired from the Water Resource Annals of Cangzhou (Xue 1994), the data after 1985 is detected from the announcements, documents of the Ministry of Water Resources, the Government of Hebei, the Water Resources Department of Hebei, the Government of Cangzhou, the Water Resources Bureau of Cangzhou. We will add the citations in the revised manuscript.

Page 9, line 19. "the environment noticeably deteriorated". How was this the case?

Response: We will add a specific statement about it, as follow:

In 2001, the cumulative subsidence was 2236 mm, with a rate of 100.45 mm/ a. The areas with subsidence larger than 500 and 800 mm are 9,717 and 3042 km2, respectively, which are 92.9% and 29.1% of the total area of Cangzhou. Besides, the interface of salt and fresh water declined around 10 m, with a maximum depth of 30 m, which threaten the fresh water in the deep aquifer (Han and Han 2006).

Page 9 & 10, has the reduction of overexploitation become a goal in itself or are earlier mentioned problems such as subsidence or salt water intrusion still an issue in the region? Are there specific quotes from governmental documents you could use to strengthen your argument?

Response: Subsidence and salt water intrusion are still issues in the region. According to the Geological environment bulletin of Hebei of 2013, the depression cone of the deep aquifer within Cangzhou is 5,551km[2].

From Page 11 onwards, How, using what method/definitions, are the more abstract general concepts as mentioned in figure 3 related to the individual, observed feedbacks as mentioned in the text? How are productive, restorative and healthy status defined? Can anything be said about what triggered the restorative force (e.g. economic motives, a change of norms and values) and/or what is meant by the steady state of the system?

Response: The emphasis level of the social productive force, can be detected from the changes in well numbers, the irrigated area with groundwater, as well as the policy for groundwater exploitation. While, the social restorative force can be detected from the changes in water saving irrigation area, as well as the policy to incent water-saving technologies. The changes in the aquifers (both shallow and deep) healthy status can be detected from the changes of the average water table depth of the shallow aquifer, and the water table depth of the depletion cone of the deep aquifer.

We think that high community sensitivity represents that humans feel the pressure of environmental deterioration, and tend to restrain human activities to restore environmental health. The change of community sensitivity is related to the change of norms and values. However, economic costs and incentives are also very important. Please refer to the response to the 2nd and 3rd comments.

Page 12, figure 4a, how are the axis defined? Is a decrease in groundwater table indicated with a positive sign? Are values calculated with regard to the groundwater table of the previous year? Has a correction been applied for water inflow (e.g. precipitation)?

Response: In the revised manuscript, we revised it as "change in shallow water table depth", and added a explanation "a negative change means a rise of the groundwater table".

Page 12, figure 4b, out of interest, how can it be that with a larger withdrawal the center of depression in 2013 is equal to 1984?

Response: The reason is that the center of the depression cone moved from urban areas to the rural area, and the average water table of the aquifer still declined. We will add a figure showing the changes in the water table depth of the aquifer Ⅲ along the cross-section from the west to the east.

[Figure]

Figure. Changes in the water table depth of the aquifer Ⅲ along the cross-section from

Page 12, figure 4c. The description of this figure is quite difficult to follow. Since two sets of data are presented in the figure, i.e. black (1976 - 2002) and blue (2003 – 2013) a discussion of these different trends would be appreciated. If individual data points are discussed as is the case now, maybe the individual years could be marked in the plot?

Response: We will mark the individual years in the plot, and add a discussion of the different trends as follow:

The dataset was divided into two periods (before and after 2002) based on the narrative of different eras, and significant changes in water user behaviour and social response to groundwater system could be found. The ratio of deep to shallow water withdrawal negatively correlated with the shallow water table depth after 2002, which is absolutely different from that before (Figure 4(c)). However, the correlation of the negative relationship is much weak with a start point before 2002 (for example, the determine coefficient is only 0.11 with data from 2002 to 2013). As shown in Fig. 2(), the deep groundwater withdrawal began to decrease slowly since 2002, while the shallow groundwater withdrawal continued to decrease rapidly as before. It indicates that people did not turn to shallow water with increasing shallow water table since the infrastructure of deep water with high quantity exploitation has already existed.

[Figure]

Page 12, line 14. "The interactions of inner Taiji : : : of blind development." On which literature is this statement based? Elinor Ostrom has, among others, done a lot of research aimed at understanding the circumstances under which overexploitation takes place.

Response: First of all, the part of sentences is trying to explain that the interactions of inner Taiji that only captures the main processes that how humans adapt to the natural variabilities and utilitze the natural resources are not enough to describe the behaviors of socio-hydrologic systems at all time and space scales. There are many cases back up the statement that without fully awareness of natural deterioration, humans development pattern will still remain blind. The most famous one is the global warming issue, even today we are not taking enough measures in coping with the overexploitation of natural resoures and over consumption of chemical fuels.

Second, E. Ostrom has done great job on socio-ecological system (SES), and based on many cases, summarized many situations that leads to overexploitation. We believe we are working on the same direction with Ostrom. While Ostrom focus on her SES framework developed from IAD framework (institutional analysis and development), our works are

mainly focus the water issue, which is quite different from other natural resources that are easily located and dividable, like woods, mines, fishery and wild animals. In future, in socio-hydrology, we believe more cases shall be studies.

Page 13, line 13, it would be helpful to see the definition of the restorative and productive force earlier in the paper, for example when introducing the Taiji Tire model.

Response: We will add the definition of the restorative and productive force when when introducing the Taiji Tire model.

In order to interpret the drivers of the reallocation of water from social-economy to the environment under the recovery stage of the pendulum swing, a concept of "environmental restorative force", which is comparative to "social productive force" was proposed (van Emmerik et al. 2014).

Page 14, line 6, "This is because at : : : protections (Elshafei et al., 2014)." On what evidence is this statement based? How is community sensitivity defined? How is it measured in the case study?

Response: In 2004, a leading group with the executive vice-mayor as the leader was established to prevent and treat the subsidence in Cangzhou. Measures for sustainable groundwater management was emphasised in the Cangzhou government work report.

Page 14, line 13. "The social productive : : : extremely costly". On what are these statements based? Is there evidence that technology and management tools are developed solely for environmental protections? Costly in relation to what (alternative)?

Response: In this paper environment is specified to the aquifers. The "new technology and management tools" denotes those for water saving. It is costly in relation to traditional groundwater exploitation. We will make the statement more clear in the revised manuscript.

Reference:
Kandasamy, J., Sounthararajah, D., Sivabalan, P., Chanan, A., Vigneswaran, S. and Sivapalan, M. (2014) Socio-hydrologic drivers of the pendulum swing between agricultural development and environmental health: a case study from Murrumbidgee River basin, Australia. Hydrology And Earth System Sciences 18(3), 1027-1041.
Sivapalan, M. (2015) Debates-Perspectives on socio-hydrology: Changing water systems and the "tyranny of small problems"-Socio-hydrology. Water Resources Research 51(6), 4795-4805.
Liu, Y., Tian, F., Hu, H. and Sivapalan, M. (2014) Socio-hydrologic perspectives of the co-evolution of humans and water in the Tarim River basin, Western China: the Taiji–Tire model. Hydrology And Earth System Sciences 18(4), 1289-1303.
Baldassarre, G.D., Viglione, A., Carr, G., Kuil, L., Yan, K., Brandimarte, L. and Blöschl, G. (2015) Debates—Perspectives on socio-hydrology: Capturing feedbacks between physical and social processes. Water Resources Research 51(6), 4770-4781.
van Emmerik, T.H.M., Li, Z., Sivapalan, M., Pande, S., Kandasamy, J., Savenije, H.H.G., Chanan, A. and Vigneswaran, S. (2014) Socio-hydrologic modeling to understand and mediate the competition for water between agriculture development and environmental health: Murrumbidgee River basin, Australia. Hydrology And Earth System Sciences 18(10), 4239-4259.

Di Baldassarre, G., Viglione, A., Carr, G., Kuil, L., Salinas, J.L. and Blöschl, G. (2013) Socio-hydrology: conceptualising human-flood interactions. Hydrology And Earth System Sciences 17(8), 3295-3303.

Elshafei, Y., Sivapalan, M., Tonts, M. and Hipsey, M.R. (2014) A prototype framework for models of socio-hydrology: identification of key feedback loops and parameterisation approach. Hydrology And Earth System Sciences 18(6), 2141-2166.

Chen, X., Wang, D., Tian, F. and Sivapalan, M. (2016) From channelization to restoration: Sociohydrologic modeling with changing community preferences in the Kissimmee River Basin, Florida. Water Resources Research 52(2), 1227-1244.

Xue, G. (1994) Water resouces annals of Cangzhou (in Chinese), Science and technology literature press, Beijing.

National Bureau of Statistics of China (2010) Hebei Compendium of Statistics 1949-1999, China Statistics Press, Beijing.

Han, Z. and Han, Y. (2006) Groundwater geo-environmental problems and control measures in Cangzhou. GroundWater 28(3), 61-64.

---

## Author Response (AR1)

**Response to Editor**

The manuscript has received two excellent reviews, both pointing to the potential interesting and important work reported, but also highlighting the weaknesses in the organization and presentation of the material. I would therefore encourage a thorough revision of the material, paying particular attention to language and presentation.

Response: The authors gratefully thank to the editor and two referees for their critical comments on our manuscript which drives us to improve the manuscript greatly. The comments and questions given by the two referees were addressed point by point. The text quoted from the revised manuscript is shown in red.

**Response to Referee #1**

General Comments

1) The objective(s) of the paper did not come across clearly. The authors state that one objective of the paper is to "analyze the co-evolution of the human-groundwater system in Cangzhou throughout history, focusing on the interactions between the social productive force and natural variability." Are there other objectives? The objective should also be introduced sooner to focus the reader. Both the "pendulum swing" concept and Taiji-Tire model are relatively new and untested ideas. The authors have not yet convinced me that they can assume these fit the case before completing the analysis. If assessing this fit is an additional paper objective, it should be made clear. Alternatively, if the paper seeks to further specify he model or test a specific aspect of it, that should also be clarified.

Response: In the revised manuscript, we tried to state the two objectives of the paper clearly: " (1) to chart the history of the groundwater utilization in Cangzhou, focusing on the dynamics of the human–groundwater interactions that lead to the "pendulum swing" in the balance point in groundwater allocations between humans and aquifer ecosystems, as well as the natural variability and social factors that contributed to it; (2) to use the Taiji-Tire model to interpret the interactions and co-evolutions of the human–groundwater system in Cangzhou. The Taiji-Tire model will be specific to the groundwater system and will be incorporated with the concepts of restorative force and community sensitivity"

In the revised manuscript, we added an introduction about the "pendulum swing" at first, and point out that the history of co-evolution of the human-groundwater system in Cangzhou fits a "pendulum swing":

" Through adaption processes, humans co-evolve with the hydrological system, resulting in a "pendulum swing" in the balance point of the human–water system (Kandasamy et al. 2014, Sivapalan 2015). Kandasamy et al. (2014) first characterized the concept of the "pendulum swing" by tracing 100 years history of the competition for water between agricultural development and environmental health in the Murrumbidgee River Basin. Similar dynamics were also found in the human–water system in the arid Tarim River Basin (Liu et al. 2014) and in human–flood interactions (Baldassarre et al. 2015). A pendulum swing can be divided into four typical stages: (1) the initial exploitation stage, which is focused exclusively on economic development; (2) the onset environmental degradation stage, which is accompanied by the introduction of remedial infrastructure; (3) the widespread environmental degradation stage, which leads to the necessity of

mitigation measures; and (4) the recovery stage, at which ultimate solutions are implemented. The pendulum swing of a human–water system can be clarified by considering all interactions within a universal socio-hydrologic framework (Kandasamy et al. 2014, Liu et al. 2014)."

We also added a more detailed introduction of the Taiji-Tire model, and clarified that the Taiji-Tire model is specified to the groundwater system, with the incorporation of the concepts of restorative force and community sensitivity.

"A socio-hydrological system contains human, hydrological, and environmental sub-systems. The Taiji-Tire model proposed by Liu et al. (2014) was first used as a framework for elucidating the complexities of socio-hydrological systems that co-evolve with direct or indirect interactions between factors from both human and water perspectives. In the model, a Taiji wheel, which is a term originating from a special concept in Chinese philosophy, is used to describe the direct human–water relationship in a specific socio-hydrological system, whereas a human–water tire is used to represent the indirect effect of external natural and social factors affecting the system. The evolution of a socio-hydrological system is driven by the interactions between two main factors, namely, natural variability and social productive force (Liu et al. 2014). Social productive force refers to the combination of all factors that enable humans to utilize resources and create better material and spiritual products. To apply the Taiji-Tire model in interpreting the drivers of water reallocation from the economy to the environment during the recovery stage of the pendulum swing, the concept of "environmental restorative force" was defined as the opposing factor of social productive force (van Emmerik et al. 2014). During the recovery stage, environmental protection actions are usually conducted because of a high social awareness for environmental risk or welfare (Di Baldassarre et al. 2013). Elshafei et al. (2014) proposed a new concept of community sensitivity to signify the social awareness for environmental welfare. Community sensitivity refers to the sensitivity of humans to the changing environment. A high community sensitivity implies that humans feel the pressure of environmental deterioration, motivating them to restrain human activities to restore environmental health. The concept of community sensitivity was also used to analyze the switching of the support between flood protection and wetland preservation in the Kissimmee River Basin, Florida (Chen et al. 2016). Above new concepts can be incorporated into the Taiji-Tire model to improve its explanatory power on a specific human–water system."

2) The methods section needs expanding as it is not clear what methods were used to develop and analyze the historical narrative. The authors use the concept of a "pendulum swing" to introduce and organize the narrative. However, it is not clear what criteria were used to determine if and when a "pendulum swing" occurs. Five eras are presented, what criteria were used to determine that a new era had begun? The Taiji-Tire model is used to frame the analysis. How was the case mapped to the Taiji-Tire? How, for example, was the spatial boundary of the internal Tire determined? And how were forces classified as productive or restorative?

Response: In the revised manuscript, we introduced the concept and definition of "pendulum swing" in the "Introduction" section, especially the different stages of it (please refer to the response to the first comment). Then, we explained the criteria used to determine if and when a "pendulum swing" occurs, and the criteria that a new era had begun as follows:

" The time series of the groundwater withdrawal from both the shallow and deep aquifers

were analyzed to detect the behaviour of water users. The average water table depth of the shallow aquifer and the water table depth of the depletion cone of the deep aquifer were used as the main indices of the groundwater system. The time series of number of wells and irrigated area (including water-saving irrigated area) were analyzed to detect the changes in infrastructure, along with the social development which were revealed by the changes in population, and grain production. The break points of the time series of the water table depth and groundwater withdrawal were determined based on trend changes. The break points were also confirmed by examining the major policies and initiatives that facilitated developments or that resulted in turning points in groundwater water management. Then, the co-evolution of the human–groundwater system was classified into several eras, which signified the points at which a "pendulum swing" occurred."

We also stated that the case mapped to the Taiji-Tire in the revised manuscript:

"Under the framework of the Taiji-Tire model, the Taiji represents the direct interactions between the groundwater utilization and the aquifers in Cangzhou, whereas the outer Tire represents all of the social and natural factors that indirectly influence the human–groundwater system. Precipitation and surface water change/variability were considered the natural variability in Cangzhou."

As for the productive and restorative forces, as mentioned in the paper, for example Fig. 4a, the restorative forces is explained as a new kind of social productive forces. The social productive force refers to the combination of all factors that help humans to utilize the resources and create better material and spiritual products that makes life better and easier. While the social productive force itself only emphasize the production but not the cost, including the direct production cost and the environmental externalities, the restorative forces refers to the specific productive forces that aiming at further increase the production by mainly lowering the environmental externalities, or, in another word, the green productivities. "The emphasis level of the social productive force is detected from the changes in the number of wells, in irrigated areas with groundwater, and in policies for groundwater exploitation. In this study, we believe that the concept of environment restorative force is misleading and should instead be regarded as a subtype of social productive force. Social restorative force can be detected from the changes in water-saving irrigation areas and in policies for creating water-saving technologies. "

3) The authors nicely demonstrate how variability in precipitation can alter the simple story of reaching a tipping point and adjusting behavior to adapt. I think this is a good contribution. However, in complex systems such as socio-hydrological systems there is great potential for multi-causality and teleconnections. In addition to groundwater levels and precipitation, were other drivers of water use behavior change considered? How were the historical narrative and data set used to focus on these drivers? Please clarify.

Response: We agree that the socio-hydrological system is multi-causality and teleconnections. The drivers of water use behavior change include precipitation, surface water resources, groundwater level, other water resources (surface inflow, water transfer from other basins, and brackish water), water-saving irrigation area, and economic conditions (the cost of groundwater abstraction and the subsidy policy). In the revised manuscript, we have aggregated these drivers from era 2 to era 5 in a new table, and give a more detailed and

concentrated historical narrative on theses drivers.

**Table 3. Changes in factors related to the community sensitivity from era 2 to era 5**

| | Precipitation mm | Surface inflow $10^6m^3$ | Alternatives ($10^6m^3$) | | Water-saving irrigation area ($10^3km^2$) [a] | Economic conditions | |
| --- | --- | --- | --- | --- | --- | --- | --- |
| | | | Brackish water | Inter-basin transfer | | Cost | Subsidy objective |
| Era 2 | 544.1 | 2144 | 0 | 0 | 26.7[a] | Low | Well drilling |
| Era 3 | 554.9 | 578 | 0 | 0 | 96.42 | Middle | No Subsidy |
| Era 4 | 391.7 | 1258 | 38.1 [b] | 41.2[c] | 212.48 | High | Water-saving |
| Era 5 | 547.2 | 1185 | 36.5 | 71.7 | 352.65 | High | Water-saving |

[a] The value at the end of each era; [b] The value of 2002, when brackish water was used at large scale; [c] The average value of 2001 and 2002;

4) While the writing did not interfere with my ability to review the manuscript there are a substantial number of grammatical errors and instances of unclear syntax. I have pointed out several, but not all, of these below. Thorough proof reading is needed before publication.

   Response: We have followed closed the suggestions made by referees, and tried to improve the use of English. The revised manuscript has been edited by a native English speaker.

Specific Comments

1) On Page 1, Line 30, define or explain what is meant by the term human forcing.

   Response: We revised this term as "social forcing (i.e., increase in water demand for agricultural development)"

2) On Page 2, Line 2, define or explain what is meant by the term salience threshold.

   Response: "salience threshold "should be "resilience thresholds" according to (Sivapalan et al. 2012)

3) In Section 2, the authors do a great job describing the hydrological and geological setting of the case. A paragraph on the governance and institutional structure of the study region would be an excellent addition here, particularly for international readers. This would help readers less familiar with Chinese governmental divisions better follow the roles of the various entities in the narrative.

   Response: We agree and added an instruction on the governance and institutional structure of the study region:

   " Cangzhou is a prefecture-level city of Hubei Province consisting of 4 county-level cities and 10 counties. Cangzhou Municipal Government abides by the provincial and national policies of water resource management, and it devises policies for the entire region. The Water Resources Bureau of Cangzhou, which is a department of Cangzhou Municipal Government, is responsible for water resource affairs and guides the Water Resources Bureau of the 10 counties and 4 county-level cities. In turn, this agency is guided by the Water Resources Department of Hebei, the Ministry of Water Resources, and the Haihe River Water Resources Commission."

4) On page 4, line 2, the authors specify the sources of hydrological, agricultural and water use data sources. However, it is not clear what the data source is for policy initiatives (Table 1) or how relevance of policy initiatives was determined.

Response: We have stated the data source in the revised manuscript:

" The data before 1985 was acquired from the Water Resource Annals of Cangzhou (Xue 1994), and the data after 1985 was obtained from the Hebei Rural Statistics Yearbook, and the Hydrology and Water Resources Investigation Bureau of Cangzhou. The data on policies and initiatives were collected from the Water Resource Annals of Cangzhou (before 1985), the announcements and documents of the Ministry of Water Resources, the Government of Hebei, the Water Resources Department of Hebei, the Government of Cangzhou, and the Water Resources Bureau of Cangzhou."

5) On page 9, line 11, in the description of the drought era (1997-2002) the authors state that well drilling "seemed to be the only choice to resist the drought." Yet, in section 3.5 they describe measures such as water licensing (1999) and irrigation efficiency improvements (1998). Why aren't these measures discussed in conjunction with the expansion of well drilling?

Response: We revised this sentence as "well drilling seemed to be the immediate strategy for addressing the drought". The measures of water licensing (1999) and irrigation efficiency improvements (1998) have been discussed in conjunction with the expansion of well drilling.

6) On page 9, section 3.5, the description of era 5 (2003-present) contains several events that occur before 2003 such as the 1999 water licensing system. Why aren't these events considered as part of era 4?

Response: In the revised manuscript, the events that occur before 2003 (the 1999 water licensing system, irrigation efficiency improvements) have been moved to era 4.

7) On page 11, figure 3, I appreciate the qualitative plotting of the level of emphasis on production and restoration. However, I would like to understand how these levels were estimated. What data sources (either quantitative or qualitative) were used? I am also unsure of the meaning of the emphasis level of "healthy status" in this context. Does healthy refer to environmental or public health? And if it refers to environmental health how does the emphasis on environmental health differ from the focus on restoration (or the restorative force)? Please clarify.

Response: The emphasis level of the social productive force, can be detected from the changes in well numbers, the irrigated area with groundwater, as well as the policy for groundwater exploitation. While, the social restorative force can be detected from the changes in water saving irrigation area, as well as the policy to incent water-saving technologies.

"healthy status" is specified as "aquifers healthy status" in this context. The changes in the healthy status of the aquifers (both shallow and deep) can be detected from the changes of the average water table depth of the shallow aquifer, and the water table depth of the depletion cone of the deep aquifer. The changes of the emphasis on the aquifers healthy status are the consequences of the restorative forces.

8) Page 12, figure 4a and page 13 figure 5a: clarify the directionality of shallow water table changes. Is a negative change a decline in groundwater levels or a decrease in the depth the ground water table?

Response: We are sorry for the misleading. In the revised manuscript, we have revised it as "change in shallow water table depth", and added a explanation "a negative change indicates a rise of the groundwater table".

9) On page 12, figure 4c conveys change in the relationship between shallow groundwater table depth and the ratio of deep to shallow groundwater. The reader needs more information to properly interpret this figure. How was this data set separated into these two groups (before and after 2002)? Was the division determined solely based on the narrative or were statistical tests used? Does any of the qualitative historical data collected aid in interpretation of this plot? What does this plot illustrate about the behavior of water users in the basin?

Response: The dataset was divided into two periods (before and after 2002) based on the narrative of different eras, and significant changes in water user behavior and social response to groundwater system could be found, as:

"significant changes in water user behavior and human response to groundwater system can be observed if the dataset was divided into two periods (before and after 2002) based on the narrative of the different eras. The ratio of deep water to shallow water withdrawal was negatively correlated with the shallow water table depth after 2002, which is absolutely different from that before (Fig. 5(c)). However, the correlation of the negative relationship is considerably weaker at a starting point before 2002 (for example, the coefficient of determination ($R^2$) is only 0.11 according to the data from 2002 to 2013). As shown in Figs. 2(g) and 2(h), the deep groundwater withdrawal began to decrease slowly since 2002, whereas the shallow groundwater withdrawal continued to decrease rapidly as before. This finding indicates that people did not turn to shallow water as the shallow water table increased, because the infrastructure of deep water with high-quantity exploitation already existed. "

10) On page 13, line 3, the authors emphasize that the social restorative force is not necessarily in opposition to the restorative force and can be considered a subset of the productive force. Is this a modification to the original Taiji-Tire model? Please clarify.

Response: First of all, please refer to our response to the 2nd comment. Secondly, it is not a modification but a supplement to the original Taiji-Tire model. Liu et al. (2014) firstly addressed the idea of Taiji-Tire model. In their work, the socio-hydrological system are seen as consequence of interactions of two general drivers, namely natural variabilities and the social productive forces (Liu et al.., 2014). When van Emmerik et al. (2014) use this Taiji-Tire idea to their work in Murrubidgee Basin, they defined a new term of natural restorative force to substitute the natural variability as the opposite factor of social productive forces. In this study, we believe that this idea of natural restorative force is confusing and should be seen as a subtype of social productive forces.

11) On page 13, lines 8 and 24 the authors make reference to the system steady state and the date in which it was broken. Please clarify what in this instance was in steady state as I am skeptical that the socio-hydrological system broadly defined was ever in steady state. Please

also describe how 1965 was identified as the end to this steady state period.

Response: We focus on the human-groundwater system within Cangzhou, and the surface water is taken as an external driver. We means that the human-groundwater system within Cangzhou was in steady state before 1965, not including the surface water. We have explained why it is defined as a steady state and how 1965 was identified as the end to this steady state period in the revised manuscript.

"Before 1965, groundwater utilization was not large scale because of technological limitations. The volume of groundwater withdrawal was small compared with that of surface water, and the irrigated area with groundwater was constrained to a small fraction of Cangzhou. According to the Taiji–Tire model, the human–groundwater relationship was weak, lacking any kind of sophisticated interactions. The human sub-system was insensitive to the groundwater sub-system, and the groundwater sub-system was unaffected by humans at a large scale. Therefore, the human–groundwater system can be considered stationary (or a steady state) without significant external drivers."

12) On page 13, figure 5b, more information is also needed to interpret this figure including how the data set was divided into two periods.

Response: The data set was divided according to the narrative of different eras. We added some information including how the data set was divided into two periods as follow:

"The groundwater withdrawal decreased continually after 2002, although the annual precipitation kept stable. The significant negative correlation between the groundwater withdrawal and annual precipitation before 2002 ($R^2$ is 0.31) (Fig. 6(b)) would significantly decrease if the period is extended to after 2002 ($R^2$ is only 0.25 when the used data is during 1976–2003). This decoupling reveals that water demand would no longer subtly vary with precipitation. Although an uncertainty still exists because the precipitation did not vary significantly during 2003–2013, the findings indicate an increased ability to mitigate climate variability."

13) On page 14, line 6 the authors discuss the accumulation of community sensitivity. What data, either quantitative or qualitative, can back up this statement? It would also be helpful to clarify what is meant by community sensitivity. In the article referenced, Elshafei et al (2014), the authors specify what community sensitivity is theorized to depend on, that would also be useful here.

Response: We added an introduction on community sensitivity at first (please refer to our response to the 1st comment), and introduced two incidents to denote the accumulation of community sensitivity: "In 2004, a leading group headed by the executive vice mayor was established to prevent and manage the subsidence in Cangzhou. Measures for sustainable groundwater management was even emphasized in the Cangzhou Government Work Report in 2004."

14) On page 14, line 12 the authors state that costly new technologies are adopted solely to protect the environment. Please note how were other motivations or causes ruled out.

Response: We have added a statement as follow:

"Given that economic development has always been the primary target of the government and society of Cangzhou, the mitigation measures for reallocating water from the economic

development to the environment, which have been implemented in Murrumbidgee River basin (Kandasamy et al. 2014), are unacceptable. Therefore, new policies are established to restrict groundwater exploitation. The development and application of water-saving technologies are strongly encouraged to protect the environment while ensuring economic development, even if such strategies are more costly than traditional groundwater exploitation."

15) On page 14, figure 6b same comment as figure 5b above.

Response: We have changed the division into 1976-1996 and 1997-2013 in the revised manuscript, and added some information as follow:

"As community sensitivity continues to increase, new technologies and management efforts would enhance the restorative forces. Supported by theses productive forces, the grain production declined slightly during the drought during 1997–2002, but it grew rapidly again with the subsequent decline in groundwater utilization after 2002 (Fig. 2(d)). Thus, the grain production was negatively correlated with the groundwater withdrawal during 1997–2013 (correlation coefficient of −0.87), which is different from that during 1976–1996 (correlation coefficient of 0.77) (Fig. 7(b))."

[Figure]

16) On page 15, the authors note that groundwater withdrawal no longer varies with precipitation. What enabled this decoupling?

Response: The extension of water-saving technologies and the construction of water-saving projects, as well as the restricted groundwater exploitation policies enabled this decoupling. Please also refer to the response to comment 12).

Technical Corrections
1) Page 1, Line28: Syntax is awkward: "Except for the social forcing, natural variability is another external forcing." Could rephrase as: "In addition to the social forcing, natural variability is an external forcing."

Response: We have revised it following the suggestion.

2) Page 2, Line 27: Correct grammatical errors: "Because that groundwater pumping from the aquifer increases obviously since middle of the 1960s, the NCP aquifer system becomes one

of the most overexploited aquifer in the world" perhaps as: "Because groundwater pumping from the aquifer has increased significantly since middle of the 1960s, the NCP aquifer system has become one of the most overexploited aquifers in the world."

Response: We have revised it following the suggestion.

3) Page 9, line 2: missing closing parentheses after the word year.

Response: We have revised it following the suggestion.

4) Page 14, line 14: replace development with developed.

Response: We have revised it following the suggestion.

5) The citation Liu et al. (2014) is missing from the references.

Response: We have added the reference.

Response to Referee #2

General comments

1) It would be helpful for the reader if concepts were clearly defined from the start. Currently, the use of concepts is mixed and includes the use of Taiji Tire model, the concept of pendulum swing (Kandasamy et al. 2014) and the concept of community sensitivity (Elshafei et al., 2014). While the pendulum swing is not defined as such, the concept of community sensitivity is introduced in the discussion, but at the same time forms a major part of the discussion. If the authors wish to use more than one concept, the reader would benefit from a more comprehensive introduction of these concepts early in the paper, possibly including their purpose and/or limitations, and how these concepts are used for the current analysis of the Cangzhou case. The latter would give the reader a clearer indication of what it can and cannot expect from the current analysis. For example, the Taiji-Tire model is merely offered as an organizing framework to represent and explain the human-water relationships (Liu et al., 2014). The conceptualization of interactions serves as a first step to a quantitative (numerical) model that can be used to explain the past and develop predictive insights (Liu et al., 2014). The pendulum swing refers to "an exclusive focus on agricultural development and food production in the initial stages and its attendant socioeconomic benefits, followed by the gradual realization of the adverse environmental impacts, subsequent efforts to mitigate these with the use of remedial measures, and ultimately concerted efforts and externally imposed solutions to restore environmental health and ecosystem services" (Kandasamy et al., 2014).

Response: We agree with the comment. In the revised manuscript, we added an introduction about the "pendulum swing" at first, and pointed out that the history of co-evolution of the human-groundwater system in Cangzhou fits a "pendulum swing". We also added a more detailed introduction of the Taiji-Tire model to clarify that the Taiji-Tire model is specified to the groundwater system. We also added an introduction of the concepts of restorative force and community sensitivity, and pointed out that they will be incorporated into the Taiji-Tire model.

Please also refer to the response to the 1st and 2nd comments of Referee #1.

2) The primary goal as defined by the authors is the interpretation of the case study using the

Taiji Tire model. It remains however unclear which methods are used to relate the observed feedbacks in the case study to the more abstract concepts in the model: what is used to distinguish endogenous from exogenous variables? How are the major drivers of the system resolved? When is a feedback considered to be productive or restorative? Currently, his seems to be dependent on your system boundary? The environmental burden seems to be partly shifted from groundwater to the surface water systems that deliver the water transfers?

Response: The inner Taiji represents the direct interacting human activities and hydrologic variables at short-term, reflecting the human-water relation. In Cangzhou, the Taiji represents the direct interactions between groundwater utilization and the water table. Therefore, the groundwater withdrawal from the shallow and deep aquifers, as well as the status of the aquifers (denoted by the shallow water table depth and water table depth of the depletion cone) are taken as the endogenous variables. The outer Tire represents all those social and natural factors that indirectly influence the system. Environmental change, especially precipitation and surface water change/variability, is an external driver of the human-groundwater system within Cangzhou. The social productive force is another external driver of the co-evolution of the human-groundwater system in Cangzhou. It should be noted that the Taiji-Tire model is specified to the groundwater system in Cangzhou. Therefore, the surface water systems is taken as a external driver. The restorative forces is explained as a new kind of social productive forces. For Cangzhou, the emphasis level of the social productive force, can be detected from the changes in well numbers, the irrigated area with groundwater, as well as the policy for groundwater exploitation. While, the social restorative force can be detected from the changes in water saving irrigation area, as well as the policy to incent water-saving technologies.

Please also refer to the response to the first and second comments of Referee #1.

3) Environmental awareness/ community sensitivity/ natural restorative force is ultimately put forward as a driver of groundwater table restoration. It is however unclear at the moment how this is inferred from the case study. The description of the case study is detailed and shows the complexity of the system including groundwater changes in both the shallow and deep water aquifer, resulting in policy developments aimed at improved groundwater management and ultimately an increase in groundwater tables. At the same time, it is mentioned that pumping costs increase due to the deeper groundwater table, land subsides (up to almost a meter), salt intrudes and additional water is available due to water transfers. How does environmental awareness relate to economic incentives or the availability of an alternative water source? If the paper would include an interpretation (qualitative or quantitative) of the strength of the various feedbacks, if would make the current conceptualization much stronger.

Response: Environmental awareness is not only related to the measurable economic costs or incentives or the availability of an alternative water source. Community sensitivity is the sensitivity of human society to the changing environment (Elshafei et al. 2014). High community sensitivity represents that humans feel the pressure of environmental deterioration, and tend to restrain human activities to restore environmental health. The increasing economic costs would result in the increasing community sensitivity. The economic incentives can be taken as a response or adaption of the increasing community sensitivity. The water transfers may reduce the community sensitivity at a local scale during a short period,

but the effects can not last for long-term if the human-groundwater relationship is not change intrinsically.

4) The text would benefit from editing and proof reading to improve its readability.

Response: We have followed the suggestions made by referees, and improve the use of English. The revised manuscript will be edited by a native English speaker.

Specific comments

Page 3, line 3. Please lengthen, since it is currently the core of the article. By which measure are feedbacks categorized? How exactly is a social productive force defined? How are main drivers distinguished among all plausible drives? Zhang et al. (2011), for example use five steps to deduct causal mechanisms in their research to link climate change and large-scale human crises.

Response: The objectives of the article have been stated clearly in the revised manuscript (Lines 29-34, page 3). We have made a detailed statement on the method about categorizing the feedbacks, defining a social productive force and distinguishing plausible drives. Please refers to the response to 2nd comment of 1# referee. And also, we agree that distinguishing the main drivers from all other plausible drivers are important. For current, however, it still need more case studies to complete. We will try to answer this question in further works.

Page 4, line 12 to 16, please check era titles with paragraph titles; not all of them match.

Response: We have revised the titles to make them consistent.

Page 4, paragraph 3.1, what are the sources used for this paragraph?

Response: The data is from the Water Resource Annals of Cangzhou (Xue 1994). We added the citation.

Page 4, paragraph 3.1, while the statements related to irrigated area argue in favor of natural variability dominating socio-hydrological change, the statements related to reservoirs, diversion projects and drainage-oriented policy currently imply that humans seems to have considerable impact already during this period. Adding a statement on the (limited?) effects of these policies on groundwater would strengthen the argument.

Response: Since we focus on human-groundwater system, so the title was revised to " Natural variability dominates the human–groundwater system . In Cangzhou, "reservoirs, diversion projects" were conducted on surface water resources, and the "drainage-oriented policy" was restrained to low lands. The groundwater sub-system was not affected by humans at a large scale. "Therefore, this era was characterized by small-scale groundwater utilization because of the low demand and technological limitations. The relationship between humans and the groundwater was weak. Humans were insensitive to groundwater change, and the groundwater sub-system was unaffected by humans at a large scale."

Page 5, figure 2, please check the figure references: subfigure d is referred to twice, resulting in a mismatch (e to i) from there on.

Response: We have revised the figure references.

Page 5, figure 2f (irrigated area), what is meant by irrigated area, is this total irrigated area or irrigated area using surface water?

Response: It means total irrigated area. We have revised it.

Page 5, figure 2, having a figure here showing the incoming water from all or the most important diversions/transfers into the region would complete the story. I can imagine that the availability of an alternative water source plays a significant role in the restoration of groundwater levels.

Response: We have added a figure for the variations of water diversion from outside the basin and the amount of brackish water. We think the new figure can be help.

[Figure]

Page 7, line 18, "Groundwater then became an important water resource for agricultural irrigation." Should this be "the most important water resource"? Given that well drilling for groundwater started a few years earlier?

Response: During this era, surface water was still more important than the groundwater. In 1982, the irrigated area with groundwater was 41.3% of the total irrigated area of Cangzhou.

Page 7 to 10, paragraphs 3.2, 3.3., 3.4, what are the sources used for these paragraphs?

Response: The data used for paragraph 3.2 is from the Water Resource Annals of Cangzhou (Xue 1994). The data of agricultural infrastructures and production used for paragraphs 3.3, 3.4, and 3.5 is from the National Bureau of Statistics of China (2010), the Hebei Rural Statistics yearbook (from 1994 to 2013). The data of annual precipitation, groundwater withdrawal from both the shallow and deep aquifers, groundwater table depth is from the Hydrology and Water Resources Investigation Bureau of Cangzhou. The data of policies and initiatives before 1985 is acquired from the Water Resource Annals of Cangzhou (Xue 1994), the data after 1985 is detected from the announcements, documents of the Ministry of Water Resources, the Government of Hebei, the Water Resources Department of Hebei, the Government of Cangzhou, the Water Resources Bureau of Cangzhou.

We have revised the statement.

Page 9, line 19. "the environment noticeably deteriorated". How was this the case?

Response: We will add a specific statement about it, as follow:

"In 2001, the cumulative subsidence was 2,236 mm, with a rate of 100.5 mm/a. The areas with subsidence larger than 500 and 800 mm are 9,717 and 3042 km$^2$, respectively, which account for 92.9% and 29.1% of the total area of Cangzhou. Moreover, the interface of saltwater and fresh water declined at approximately 10 m, with a maximum depth of 30 m, threatening the

fresh water in the deep aquifer (Han and Han 2006)."

Page 9 & 10, has the reduction of overexploitation become a goal in itself or are earlier mentioned problems such as subsidence or salt water intrusion still an issue in the region? Are there specific quotes from governmental documents you could use to strengthen your argument?

Response: " Nonetheless, the aquifers still suffer from serious environmental problems. The average water table of the aquifer is still deep although the depth in 2013 is equal to that in 1984, as the center of the depression cone moved from urban areas to the rural areas (Fig. 3). According to the Geological Environment Bulletin of Hebei of 2013, the depression cone of the deep aquifer in Cangzhou is 5,551 km$^2$."

From Page 11 onwards, How, using what method/definitions, are the more abstract general concepts as mentioned in figure 3 related to the individual, observed feedbacks as mentioned in the text? How are productive, restorative and healthy status defined? Can anything be said about what triggered the restorative force (e.g. economic motives, a change of norms and values) and/or what is meant by the steady state of the system?

Response: The restorative forces is explained as a new kind of social productive forces. For Cangzhou, the emphasis level of the social productive force, can be detected from the changes in well numbers, the irrigated area with groundwater, as well as the policy for groundwater exploitation. While, the social restorative force can be detected from the changes in water saving irrigation area, as well as the policy to incent water-saving technologies. The emphasis level of the social productive force, can be detected from the changes in well numbers, the irrigated area with groundwater, as well as the policy for groundwater exploitation. While, the social restorative force can be detected from the changes in water saving irrigation area, as well as the policy to incent water-saving technologies.

"healthy status" is referred to the aquifers (both shallow and deep) healthy status. The changes in the aquifers healthy status can be detected from the changes of the average water table depth of the shallow aquifer, and the water table depth of the depletion cone of the deep aquifer.

We think that the community sensitivity toward aquifer environment has triggered the restorative force, and caused new actions toward environmental protection. High community sensitivity represents that humans feel the pressure of environmental deterioration, and tend to restrain human activities to restore environmental health. The change of community sensitivity is related to the change of norms and values. However, economic costs and incentives are also very important. Please also refer to the response to the 2nd and 3rd comments.

Page 12, figure 4a, how are the axis defined? Is a decrease in groundwater table indicated with a positive sign? Are values calculated with regard to the groundwater table of the previous year? Has a correction been applied for water inflow (e.g. precipitation)?

Response: In the revised manuscript, we revised it as "change in shallow water table depth", and added a explanation "a negative change means a rise of the groundwater table".

Page 12, figure 4b, out of interest, how can it be that with a larger withdrawal the center of depression in 2013 is equal to 1984?

Response: The reason is that the center of the depression cone moved from urban areas to the rural area, and the average water table of the aquifer still declined. We added a figure showing the changes in the water table depth of the aquifer Ⅲ along the cross-section from the west to the east.

[Figure]

Figure. Changes in the water table depth of the aquifer Ⅲ along the cross-section from the west to the east.

Page 12, figure 4c. The description of this figure is quite difficult to follow. Since two sets of data are presented in the figure, i.e. black (1976 - 2002) and blue (2003 – 2013) a discussion of these different trends would be appreciated. If individual data points are discussed as is the case now, maybe the individual years could be marked in the plot?

Response: We have marked the individual years in the plot, and added a discussion of the different trends. Please refer to the responses to the 9th comment of Referee #1.

[Figure]

Page 12, line 14. "The interactions of inner Taiji : : : of blind development." On which literature is this statement based? Elinor Ostrom has, among others, done a lot of research aimed at understanding the circumstances under which overexploitation takes place.

Response: First of all, the part of sentences is trying to explain that the interactions of inner Taiji that only captures the main processes that how humans adapt to the natural variabilities and utilize the natural resources are not enough to describe the behaviors of socio-hydrologic systems at all time and space scales. There are many cases back up the

statement that without fully awareness of natural deterioration, humans development pattern will still remain blind. The most famous one is the global warming issue, even today we are not taking enough measures in coping with the overexploitation of natural resoures and over consumption of chemical fuels.

Second, E. Ostrom has done great job on socio-ecological system (SES), and based on many cases, summarized many situations that leads to overexploitation. We believe we are working on the same direction with Ostrom. While Ostrom focus on her SES framework developed from IAD framework (institutional analysis and development), our works are mainly focus the water issue, which is quite different from other natural resources that are easily located and dividable, like woods, mines, fishery and wild animals. In future, in socio-hydrology, we believe more cases shall be studies.

Page 13, line 13, it would be helpful to see the definition of the restorative and productive force earlier in the paper, for example when introducing the Taiji Tire model.

Response: We have added the definition of the restorative and productive force when introducing the Taiji Tire model. Please also refer to the response to the 2nd comment.

Page 14, line 6, "This is because at : : : protections (Elshafei et al., 2014)." On what evidence is this statement based? How is community sensitivity defined? How is it measured in the case study?

Response: This statement is based on these incidents: "In 2004, a leading group headed by the executive vice mayor was established to prevent and manage the subsidence in Cangzhou. Measures for sustainable groundwater management was emphasized in the Cangzhou Government Work Report in 2004."

The definition of community sensitivity can be refer to the response to the 3rd comment. In this case study, it is indicated by the measures taken by the public and policymakers to modify the existing development pattern.

Page 14, line 13. "The social productive : : : extremely costly". On what are these statements based? Is there evidence that technology and management tools are developed solely for environmental protections? Costly in relation to what (alternative)?

Response: We added the basis in the revised manuscript: "Given that economic development has always been the primary target of the government and society of Cangzhou, the mitigation measures for reallocating water from the economic development to the environment, which have been implemented in Murrumbidgee River basin (Kandasamy et al. 2014), are unacceptable. Therefore, new policies are established to restrict groundwater exploitation. The development and application of water-saving technologies are strongly encouraged to protect the environment while ensuring economic development, even if such strategies are more costly than traditional groundwater exploitation. As shown in Fig. 2(f), the area of water-saving irrigation has increased rapidly."

The "new technology and management tools" denotes those for water saving. It is costly in relation to traditional groundwater exploitation. We have made the statement more clear in the revised manuscript.

Reference:

[revised manuscript text omitted]

---

## Author Response (AR3)

**Response to the reviewer**

The authors gratefully thank to the editor and referee for their critical comments on our manuscript which drives us to improve the manuscript greatly. The comments and questions were addressed point by point. In addition, we have discussed the different roles of the two droughts (occurred in 1965 and during 1997-2002) during the evolution of the human–groundwater system.

1. The primary issue with the paper in its current state is lack of specificity of the methodology. The paper has both a compelling aim (understanding the co-evolution of groundwater and irrigation systems) and an innovative approach (pairing a pendulum swing narrative of the case and statistical analysis of hydrological variables). However, the paper as is does not describe the methodology in sufficient detail for other researchers to replicate or build upon this work. Additionally, the data used to infer changes in community sensitivity is not clear. Changes in community sensitivity are central to the interpretation of observations and case events. Therefore, pointing readers to supporting data is critical. Specifically, I ask the authors to address the following points made in the first round on comments:

   Response: We have revised the methods section, and tried to describe the methodology in sufficient detail. Firstly, the data used is described in more detail. Secondly, we describe how the pendulum swing narrative was developed also in more detail. Then, the data and method to detect the changes in community sensitivity are stated. Especially, we revised the table of the major events from two aspects: divided the major events to the four eras, and added a column to list the major events related with the environment. Finally, we describe how the Taiji-Tire model is used to map the case of Cangzhou.

2. The fact that actions are taken in response to changing community sensitivity is a central point in the paper. Please describe the sources of data, either quantitative or qualitative, used to estimate changes in community sensitivity and explain how it was used to assess change.

   Response: We have described the community sensitivity as follows:

   "The community sensitivity of the human-groundwater system is specified to the social attitude against groundwater exploitation and social awareness on the aquifers environmental crises. In Cangzhou, as well as in Hebei, community concerns about water resource crisis are represented by, and always conveyed to the policy makers through the Hydraulic Engineering Society, which is a community organization composed of individuals, institutions, and other stakeholders related with the affairs of water resources, and aims to promote sustainable water resources management. In order to detect the changes of the community sensitivity, the attitude and awareness of the Cangzhou Hydraulic Engineering Society, and the Hebei Hydraulic Engineering Society are carefully reviewed."

   We have added an explanation about the increasing community sensitivity around

1980s:

"In 1983, the Cangzhou Hydraulic Engineering Society, which was set up in 1979 as the response to public concerns on water problems, submitted an appeal document entitled "Appeal for the Rational Exploitation of Water Resources" to the government. In this appeal, proposals for comprehensive water resource management were proposed. The appeal revealed that humans felt the pressure of the aquifer deterioration, and can be regarded as an increasing community sensitivity."

We also have added an explanation about the increasing community sensitivity during Era 4:

"Meanwhile, the concerns for the environment risk was accumulating with the aquifer degradation. In 1997, the Irrigation and Drainage Group of Hebei Hydraulic Engineering Society organized a meeting on water saving agriculture, and pointed out the environmental problems related to the deterioration of the aquifer, and appealed humans to conduct high efficiency water saving. The Disaster Prevention and Mitigation Group of Hebei Hydraulic Engineering Society, which was established in 1996, committed themselves to change the idea of drought relief from well drilling to integrated measures. In 2002, they suggested the government and water conservancy bureau to take water saving and water conservation as a priority, and to reduce groundwater withdrawal. Above events revealed an obvious increase of the community sensitivity."

3. In the methods section:

(1) Explain the criteria used to select the factors related to community sensitivity included Table 3.

Response: We intend to use Table 3 to describe the outer drivers of the system from Era 2 to Era 5. We are sorry for the misleading. Please also refer to the response to the second comment.

(2) Explain how the pendulum swing narrative (era descriptions) was developed. What criteria were used to judge where one error begins and ends? What criteria were used to decide which events were relevant to include in the era descriptions?

Response: We have revised Table 1 for a better display of the events, and explained how the pendulum swing narrative was developed as follows:

"The history of the pendulum swing in the balance point of the human–groundwater system is traced through a review of the literature on the evolution of the human-groundwater system. Five distinct eras, a pre-era and four eras following the typical narrative of a pendulum swing (exploitation, degradation and restoration, drought-triggered deterioration, and returning to the balance) are divided by distinguishing the events that can be regarded as tipping points. There are two criteria to judge the tipping points: (1) groundwater utilization pattern is significantly different; (2) groundwater table of the shallow and (or) deep aquifers is significantly changed before and after the events. In addition, the division is validated through the statistical analysis of the quantitative data of the groundwater table depth and groundwater utilizations. The events in terms of the relative emphasis placed on social

development, that significantly affect the groundwater utilization or aquifers environmental health are chosen to conduct the narrative (Table 1).

The drought occurred in 1965, which triggered the groundwater utilization in Cangzhou, was chosen as the beginning of the exploitation era. The period before is regarded as the pre-era, of which the human-groundwater system was dominated by natural variability. During the exploitation era, the groundwater withdrawal began to increase, and the water table began to decline. But the trends ceased temporarily in 1983, when the Cangzhou Hydraulic Engineering Society submitted an appeal to government for integrated management of groundwater resources. After that, comprehensive management measures were implemented to address the groundwater crisis, and the deterioration was mitigated to a certain extent. Therefore, this event is chosen as the beginning of Era 3. During the Era 3, the increasing of groundwater utilization and the declines of water table began to slow down. However, the mitigation was terminated by a persistent drought lasted from 1997 to 2002, which is regarded as Era 4, when a pulse in groundwater utilization and aquifer depletion occurred. After the drought, the system has been returning to the balance with decreasing groundwater utilization and rising water table, which is regarded as Era 5."

(3) Explain how the Taiji-Tire model is used to interpret the case events and interactions. The authors explain the theory behind the model sufficiently but need to describe the logic or criteria used to map the observations to the model components.

Response: We have explained how the Taiji-Tire model is used to interpret the case events and interactions. as follows:

"Under the framework of the Taiji-Tire model, the Taiji is specified to the direct interactions between the groundwater utilization and the aquifers in Cangzhou. The status of the shallow aquifer is represented by the water table depth, while the status of the deep aquifer is represented by the depth of the depression cone center. The influences of humans on groundwater are detected from the relationship between changes in shallow groundwater table and the water withdraw, and the co-evolution of the depth of the depression cone center of deep aquifer with water withdraw from 1976 to 2013. The feedbacks of changes in groundwater sub-system on humans were detected from the relationship between groundwater use regime and the shallow water table depth. The outer Tire represents all of the social and natural factors that indirectly influence the human–groundwater system. The time series of annual precipitation and surface inflow were used to represent the natural variability in Cangzhou. The impacts of the natural variability on the groundwater sub-system and human sub-system were detected from the changes in groundwater table depth and withdrawals corresponding to the changes in surface water inflow and annual precipitation, 
[revised manuscript text omitted]

---

## Author Response (AR4)

**Response to the reviewer**

The authors gratefully thank to the editor and referee for their critical comments on our manuscript which drives us to improve the manuscript greatly. The comments and questions were addressed point by point.

I thank the authors for the revisions made to clarify the methodology and the data used to assess community sensitivity. My major concerns have been addressed and I believe the article is ready for publication once two technical issues have been addressed.

The authors clarify that the actions of the Hydraulic Engineering Society are used as a proxy for measuring community sensitivity. Please note why the Hydraulic Engineering Society is an appropriate proxy (i.e. largest non-governmental organization working on groundwater issues, most diverse membership, etc.).

Response: We add the two reasons in the revised manuscript:

"In Cangzhou, as well as in Hebei, community concerns about water resource crisis are represented by, and always conveyed to the policy makers through the Hydraulic Engineering Society mainly because of two reasons. First, the Hydraulic Engineering Society is the largest non-governmental organization related with the affairs of water resources in corresponding region, and aims to promote sustainable water resources management. Besides, the Hydraulic Engineering Society has the most diverse membership, including individuals, enterprises, public institution, colleges and other stakeholders. Groundwater issues are among the most concerned ones of water resources management in Cangzhou. In order to detect the changes of the community sensitivity, the attitude and awareness toward groundwater issues of the Cangzhou Hydraulic Engineering Society, and the Hebei Hydraulic Engineering Society are carefully reviewed."

Table two notes that the shallow groundwater table declined 0.39 m/10 years in Era 2, while the text states that the groundwater table declined 0.39 m per year. Please reconcile these two rates.

Response: The rate should be 0.39 m per year. The unit of the trends in deep groundwater table is also "m/yr". We have revised the units in Table 2.

[revised manuscript text omitted]